# Levels of Small Extracellular Vesicles Containing hERG-1 and Hsp47 as Potential Biomarkers for Cardiovascular Diseases

**DOI:** 10.3390/ijms25094913

**Published:** 2024-04-30

**Authors:** Luis A. Osorio, Mauricio Lozano, Paola Soto, Viviana Moreno-Hidalgo, Angely Arévalo-Gil, Angie Ramírez-Balaguera, Daniel Hevia, Jorge Cifuentes, Yessia Hidalgo, Francisca Alcayaga-Miranda, Consuelo Pasten, Danna Morales, Diego Varela, Cinthya Urquidi, Andrés Iturriaga, Alejandra Rivera-Palma, Ricardo Larrea-Gómez, Carlos E. Irarrázabal

**Affiliations:** 1Laboratory of Molecular and Integrative Physiology, Physiology Program, Centro de Investigación e Innovación Biomédica (CiiB), Universidad de los Andes, Santiago 7620001, Chile; luis.osorio@miuandes.cl (L.A.O.); mpasten@uandes.cl (C.P.); 2Laboratory of Nano-Regenerative Medicine, Center of Interventional Medicine for Precision and Advanced Cellular Therapy (IMPACT), Centro de Investigación e Innovación Biomédica (CiiB), Universidad de los Andes, Santiago 7620001, Chile; 3Faculty of Medicine, Universidad de los Andes, Santiago 7620001, Chile; 4Faculty of Medicine, Universidad de Chile, Santiago 8380453, Chile; 5Department of Epidemiology and Health Studies, Facultad de Medicina, Universidad de los Andes, Santiago 7620001, Chile; 6Departamento de Matemática y Ciencia de la Computación, Facultad de Ciencia, Universidad de Santiago de Chile, Santiago 9170020, Chile; 7Research Unit, Clínica Dávila, Santiago 8431657, Chile; 8Cardiovascular Department, Clínica Dávila, Santiago 8431657, Chile; ricardolarrea.externo@davila.cl

**Keywords:** human ether-a-go-go-related gene 1, heat shock protein 47, small extracellular vesicles, cardiovascular disease

## Abstract

The diagnosis of cardiovascular disease (CVD) is still limited. Therefore, this study demonstrates the presence of human ether-a-go-go-related gene 1 (hERG1) and heat shock protein 47 (Hsp47) on the surface of small extracellular vesicles (sEVs) in human peripheral blood and their association with CVD. In this research, 20 individuals with heart failure and 26 participants subjected to cardiac stress tests were enrolled. The associations between hERG1 and/or Hsp47 in sEVs and CVD were established using Western blot, flow cytometry, electron microscopy, ELISA, and nanoparticle tracking analysis. The results show that hERG1 and Hsp47 were present in sEV membranes, extravesicularly exposing the sequences ^430^AFLLKETEEGPPATE^445^ for hERG1 and ^169^ALQSINEWAAQTT- DGKLPEVTKDVERTD^196^ for Hsp47. In addition, upon exposure to hypoxia, rat primary cardiomyocytes released sEVs into the media, and human cardiomyocytes in culture also released sEVs containing hERG1 (EV-hERG1) and/or Hsp47 (EV-Hsp47). Moreover, the levels of sEVs increased in the blood when cardiac ischemia was induced during the stress test, as well as the concentrations of EV-hERG1 and EV-Hsp47. Additionally, the plasma levels of EV-hERG1 and EV-Hsp47 decreased in patients with decompensated heart failure (DHF). Our data provide the first evidence that hERG1 and Hsp47 are present in the membranes of sEVs derived from the human cardiomyocyte cell line, and also in those isolated from human peripheral blood. Total sEVs, EV-hERG1, and EV-Hsp47 may be explored as biomarkers for heart diseases such as heart failure and cardiac ischemia.

## 1. Introduction

Cardiovascular diseases (CVDs) are the leading cause of death worldwide, and they are associated with common risk factors such as smoking, reduced physical activity, an unhealthy diet, overweightness and obesity, high cholesterol and blood pressure, and anomalous glycemia [1,2]. The World Health Organization (WHO) estimates that 17.9 million people die each year because of CVDs, an estimated 32% of all deaths worldwide [3]. Cardiac ischemia, also referred to as coronary heart disease (CHD) and coronary artery disease (CAD), is defined as an inadequate blood supply to the heart due to dysfunction of the heart’s coronary arteries. Ischemia is a self-propagating process that irreversibly deteriorates cardiac function and negatively impacts prognosis [4]. In addition, patients with CHD have an elevated risk of experiencing sudden acute myocardial infarction (AMI) and death [5].

Currently, numerous diagnostic tests are available to diagnose CHD, which differ in terms of their sensitivity, specificity, cost, and accessibility to patients, and some of them are highly invasive. The most widely used is the cardiac stress test, which evaluates how the heart works during physical activity. It is quick and noninvasive. More than 10 million cardiac stress tests are performed annually in the United States. However, in a meta-analysis, it was established that the cardiac stress test has limited sensitivity (average: 67%; 54–78%) and specificity (average: 46%; 30–64%), with a serious problem concerning differences related to the sex of the patient (positive predictive value: women 22% and men 48%) [6], leaving women with lower odds of a good diagnosis and treatment [6]. In addition, cardiac stress tests require a cardiologist trained in the use of cardiology equipment in a laboratory setting. There are other tests that have greater sensitivity and specificity, such as the echocardiogram, coronary angiogram, and nuclear cardiac stress test. However, it should be noted that these tests must be performed at medical centers with specialized health personnel (cardiologists, radiologists, nurses, etc.), nuclear medicine units, cardiology departments, imaging laboratories, and high-cost equipment [7]. Additionally, some tests require that the patient not present other pathologies, such as allergies to contrast media or other chronic diseases (for example, chronic kidney and liver diseases) [8,9]. Thus, the diagnosis of coronary heart disease is limited and, in some cases, leads to inadequate treatment.

Heart failure (HF), also known as congestive heart failure, is a condition that develops when the heart does not pump enough blood for the needs of the body. HF is the leading cause of premature death worldwide in patients with established CVD, regardless of the specific clinical phenotype [10]. The International Heart Societies report strongly agrees with the use of natriuretic peptides (NPs) in diagnosing, predicting, and managing HF. Thus far, the B-type natriuretic peptide (BNP) and N-terminal proBNP (NT-proBNP) are the gold standard biomarkers for the diagnosis and prognosis of HF. The recent 2022 Heart Failure Guidelines provide alternative biomarkers (galectin-3 and soluble suppressor tumorigenicity-2) for risk stratification and prediction of outcomes [11]. In addition, cardiac troponin I (cTnI) and troponin T (cTnT) are the accepted biomarkers for the detection of myocardial necrosis and the diagnosis of AMI [12]. Recently, high-sensitivity assays have been developed to detect cTn concentrations measurable in >50% of normal healthy individuals. The high-sensitivity cTn test is being used to explore different CVDs [13]. However, given that some patients with HF do not respond to the treatment, the development of new biomarkers to evaluate the evolution from a compensated to decompensated HF condition will open new therapeutic opportunities to better treat this condition.

Small extracellular vesicles (sEVs) are membrane-bound vesicles secreted by cells into the extracellular space [14]. sEVs are recognized as potent vehicles for intercellular communication because of their capacity to transfer proteins, lipids, and nucleic acids, thereby influencing various physiological and pathological functions [15]. The presence of circulating sEVs has been described in different cardiovascular diseases [16], and on the basis of their relevance and potential, they are being considered as potential biomarkers [17]. The nanoparticle tracking analysis (NTA) methodology with NanoSight equipment is currently used to determine the concentrations and sizes of extracellular vesicles in a biological fluid. This technique allows us to visualize and quantify particles in real time from 50 to 1000 nm in size [18]. It is also used to measure the concentrations and sizes of sEVs in blood samples [19]. In addition, NTA can even be combined with fluorescence (NTA-F) using antibodies conjugated with a fluorophore to characterize specific proteins in vesicle membranes [18,19,20]. Thus, proteins in vesicular membranes can be quantified using an innovative strategy that quantifies sEVs with specific markers.

In vitro studies have described the potential protective role of sEVs in cardiomyocytes by activating membrane receptors that trigger signaling pathways involved in cell protection [21,22]. Furthermore, an in vivo study determined the cardioprotective role of sEVs containing heat shock protein 70 (Hsp70) in the vesicular membrane (EV-Hsp70). As a result, EV-Hsp70 binds to toll-like receptor 4 (TLR4), activating signaling pathways involved in cardioprotective effects through Hsp27 [23,24].

In addition, Hsp47 (also named Serpin H1) is expressed in the heart during stress episodes and when it is injured [25,26], and this protein is considered to be a stress-inducible collagen-specific molecular chaperone involved in the processing and secretion of procollagen. In addition, the voltage-dependent potassium channel a-subunit KV11.1 (also named the human ether-a-go-go-related gene 1 (hERG1)) is essential for normal electrical activity in the heart, regulating the cardiac action potential [27,28]. The mutations in the hERG1 lead to the development of long QT syndrome (LQTS), a cardiac repolarization disorder that predisposes affected individuals to arrhythmia, i.e., rapid, irregular heartbeats that can lead to fainting and sudden death [29]. 

The main objective of the present study was to characterize the presence of hERG1 and Hsp47 associated with the membranes of extracellular vesicles in blood samples as potential biomarkers of cardiovascular diseases.

## 2. Results

### 2.1. hERG1 and Hsp47 Are Expressed on the Surfaces of sEVs

We studied the abundance of sEV proteins via Western blot using an enriched fraction of nanoparticles from human peripheral blood. We found that (i) some of them were highly expressed in the heart, and that (ii) they were associated with the membranes of the small extracellular vesicles. The platelet-free plasma (PFP) was prepared from human blood samples within the first hour of the blood sampling. Then, the extracellular vesicles were enriched by ultracentrifugation in a pooled sample from the participants with a positive diagnosis of cardiac ischemia (verified by coronary angiogram), as described in Table 1. Using the anti-hERG1 antibody, which recognizes the extracellular domain (^430^AFLLKETEEGPAPATE^445^) of hERG1, we found in the Western blot analysis that hERG1 was present in the sEVs as identified by CD9 (Figure 1A), and not for intracellular proteins (calnexin or Grp94; Appendix A). Then, the expression of hERG1 in whole sEVs obtained from a pool of samples from participants with a positive diagnosis of cardiac ischemia was studied via flow cytometry with the same anti-hERG1 antibody. Using synthetic beads to capture whole sEVs, we did not observe a positive signal in the 12,000× *g* centrifugation pellet (sEV-free fraction), even when the anti-hERG1 antibody was included in the reaction (Figure 1B, see the red signal). However, a positive signal in the 110,000× *g* centrifugation pellet (sEV-enriched fraction) was observed with anti-hERG1. Finally, we studied the capability of detecting hERG1 directly from blood samples using a flow cytometer, avoiding the ultracentrifugation process. Interestingly, we did not observe a positive signal when PFP was used directly in the flow cytometer measurements. However, when the PFP was preclearing from the protein components, using a small column of size exclusion chromatography (SEC), a positive signal was observed concomitant with a positive signal for sEV markers CD9, CD63, and CD81 (Appendix A). Transmission electron microscopy (TEM) was also used to show hERG1 association with sEVs (Appendix A). Together, our data suggest that the ^430^AFLLKETEEGPAPATE^445^ epitope of hERG1 is present on the surfaces of sEVs obtained from human peripheral blood.

Hsp47 was also found to be expressed on the surfaces of the extracellular vesicles. Previously, it was documented that the stress-inducible heat shock protein 70 (Hsp70) was expressed on the surface of tumor cells and released into the blood via small extracellular vesicles [30]. The sequence TKD (TKDNNLLGRFELSG) was identified as being associated with the extravesicular domain. Thus, we explored the TKD sequence in the Hsp47 protein and found that the sequence “^189^TKDVERTDGAL^200^” was present in the protein. Using a commercial antibody against human Hsp47 that recognizes the mentioned sequence, the expression of Hsp47 in sEVs was studied. The sEVs, purified by ultracentrifugation (from pooled PFP of participants positive for cardiac ischemia), showed that Hsp47 was present in the sEVs as identified by CD9 in the Western blot analysis (Figure 1A). Flow cytometry experiments produced a negative signal for the sEV-free 12,000× *g* centrifugation pellet (Figure 1B), but a positive signal for the 110,000× *g* centrifugation sEV pellet (Figure 1D), suggesting that the sequence “TKDVERTDGALL” of Hsp47 is exposed to the extravesicular domain of sEVs. Transmission electron microscopy (TEM) was used to indicate sEV surface localization of the Hsp47 epitope (ALQSINEWAAQTTDGKLPEVTKDVERTD) (Appendix A).

### 2.2. Effect of Hypoxia on the Secretion of sEVs Containing hERG1 and Hsp47 Proteins Derived from Cardiomyocyte Cell Culture

Using a primary cell culture of cardiomyocytes from rat neonates, we studied the effect of hypoxia on sEVs secretion using a time–course curve. We measured the levels of sEVs by nanoparticle tracking analysis (NTA) in the NanoSight instrument, directly from the culture media (without ultracentrifugation). The results showed that after 15 min of exposure to 1% oxygen, the cardiomyocytes significantly increased the release of sEVs compared with 21% oxygen, reaching a peak at 30 min, and then decreasing gradually over 8 h of exposure to hypoxia, which still resulted in higher levels of release than with 21% oxygen (Figure 2A). Thus, the cardiomyocytes released sEVs due to hypoxia as quickly as 15 min after oxygen deprivation. Using the human cardiomyocyte (AC-16) cell line (Appendix A) and an in-house ELISA, we studied the effect of hypoxia on the sEVs containing hERG-1 or Hsp47 proteins. The ELISA used to detect the hERG1 associated with the sEVs consists of capturing sEVs with the anti-hERG1 antibody, and conducting a reading with the anti-CD81 HRP-conjugated antibody. On the other hand, the ELISA used to detect the Hsp47 associated with the sEVs consists of capturing sEVs with the anti-Hsp47 antibody, and reading it with the anti-CD63 HRP-conjugated antibody. Both types of ELISA read the protein present in the extravesicular domain of the whole sEVs. For 1 h, we exposed the cardiomyocytes to normoxia (21% O_2_) or hypoxia (1% O_2_) in an FBS-depleted culture medium. The sEV concentration was normalized with the total protein amount present in each plate. Thus, we observed that hypoxia (1% O_2_) significantly induced the level of sEVs containing the hERG1/CD81 (Figure 2B) and Hsp47/CD63 (Figure 2C) configurations, in comparison to the normoxia (21% O_2_) condition. These data suggest that cardiomyocyte cell line response to hypoxia is to release sEVs containing hERG1 and Hsp47, suggesting a cellular phenomenon associated with the hypoxia response.

### 2.3. Blood Levels of sEVs Containing hERG1 and Hsp47 Proteins in the Stress Test

Considering the preliminary in vitro data, we decided to study sEV concentrations in human blood samples from the cardiac stress test participants. The stress test causes the heart to pump harder and faster, so that the test can detect blood flow problems in the heart. Thus, cardiac ischemia is caused and diagnosed during stress tests, establishing a coronary artery disease condition.

A small group of participants were enrolled in the stress test via the cardiology unit of Clínica Dávila (Table 1). No significant differences were observed in sex, age, arterial hypertension (AHT), type 2 diabetes mellitus (T2DM), dyslipidemia, smoking, or acute myocardial infarction (AMI). We sampled healthy participants under resting conditions (i.e., without exercise).

We studied sEV concentrations in healthy and stress test participants with negative or positive cardiac ischemia diagnoses, using the NTA technique after a 15-min test. The healthy controls (without stress test) and negative cardiac ischemic participants had similar concentrations of sEVs in their blood. Interestingly, the participants with a positive cardiac ischemia diagnosis had a significantly higher sEV concentration than those with a negative cardiac ischemia diagnosis, 15 min after the test finished (Figure 3A), suggesting that cardiac ischemia increases the concentration of sEVs in the blood. Interestingly, one participant with a negative cardiac ischemia diagnosis had elevated levels of sEVs in their blood during the stress test (Figure 3A, red arrow). This participant has a long history of drug abuse, and these conditions may be associated with a cardiac condition and elevated sEV levels in their blood.

To study the concentrations of sEVs containing hERG1 or Hsp47 in the blood samples, we used an experimental approach of the NTA platform, with either anti-hERG1 or anti-Hsp47 antibodies, and a fluorescent secondary antibody (NTA-Fluorescence) for detection. We previously determined the optimized number of sEVs for the assay. On the basis of the preliminary data, we used 1 × 10^9^ sEVs as the standard amount for the NTA fluorescence analysis. A small volume of PFP (5–100 μL) was used to obtain the total amount of 1 × 10^9^ sEVs. The PFP samples were diluted to 0.5 mL with DPBS, and incubated for 1 h with the respective primary antibody (anti-hERG1 or anti-Hsp47). Then, an additional 1-h incubation period was conducted with the secondary antibody conjugated with Alexa Fluor 532. Finally, the antibodies were not washed out, because the antibody’s molecule size was approximately 10 nm. Thus, nanoparticles smaller than 50 nm were not considered in the analysis.

The same groups of individuals, as described in Table 1, were analyzed via NTA with fluorescence. Concentrations of sEVs containing hERG1 (EV-hERG1) were detected at different nanoparticle sizes (Figure 3B). Interestingly, the group with a positive cardiac ischemia diagnosis had a significantly higher average concentration of EV-hERG1 than those with a negative cardiac ischemia diagnosis (Figure 3B). In addition, EV-hERG1 detected between 60 and 120 nm was expressed more in the cardiac-ischemia-positive group. In the same group of participants, we studied the concentration levels of sEVs containing Hsp47 (EV-Hsp47) in PFP. The group classified as cardiac ischemia had a significantly higher average concentration of EV-Hsp47 than the cardiac-ischemia-negative participants (Figure 3C). In addition, EV-Hsp47 detected between 120 and 300 nm was expressed more in the cardiac-ischemia-positive group.

### 2.4. Effect of Chronic Heart Failure on sEVs, EV-hERG1, and EV-Hsp47 Levels in Blood Samples

We became interested in measuring the total sEVs, EV-hERG1, and EV-Hsp47 levels in another group of cardiovascular patients at the cardiology unit of Clínica Dávila with a diagnosis of compensated heart failure (CHF) and decompensated heart failure (DHF). The patients were enrolled, considering the inclusion and exclusion criteria mentioned in Section 2. The group included 10 participants with a diagnosis of CHF, and 10 participants with DHF. The data showed no significant differences in sex, age, diastolic blood pressure (DBP), heart rate, arterial hypertension (AHT), dyslipidemia, diabetes mellitus (DM), chronic kidney disease, smoking, or implantable cardioverter–defibrillator (ICD) for the enrolled participants. However, we observed a slight but significant increase (*p* = 0.047) in systolic blood pressure (SBP) in patients with DHF (146.18 ± 32.14) compared to CHF (119.78 ± 22.77) (Table 2).

We characterized the sEVs, EV-hERG1, and EV-Hsp47 in blood samples from patients with CHF and DHF. In this way, the sEVs were enriched by ultracentrifugation from the PFP of each participant, and the hERG1, Hsp47, and CD9 proteins were characterized by Western blot (Figure 4).

In addition, the characterizations of EV-hERG1 and EV-Hsp47 in the samples from participants with CHF (n = 10) and DHF (n = 10) were conducted using a flow cytometer (a representative image is shown in Figure 5A,B). Analyzing the results of the flow cytometry, we found that both groups of participants with CHF and DHF showed detectable levels of EV-hERG1 and EV-Hsp47. Interestingly, using a flow cytometer, we found that the EV-Hsp47/EV-hERG1 ratio was significantly lower in patients with DHF compared to CHF (Figure 5C), meaning a greater amount of EV-hERG1 than EV-Hsp47. Therefore, both kinds of proteins were detected in the sEVs from the blood of participants with compensated and decompensated heart failure (Figure 4 and Figure 5).

### 2.5. Levels of sEVs Containing hERG1 and Hsp47 Decreased in the Blood of Participants with Decompensated Heart Failure

Using NTA-associated fluorescence on platelet-free plasma from each participant, we found that all the healthy (n = 10), CHF (n = 10), and DHF (n = 10) participants had detectable levels of EV-hERG1 and EV-Hsp47 in their PFP samples, with similar distributions in the sizes of the sEVs (Figure 6A). The levels of sEVs increased in the CHF group compared with the healthy group. Additionally, it was determined that participants with DHF had a significantly lower concentration of sEVs than those with CHF or the healthy group (Figure 6B). In addition, the level of EV-hERG1 increased in CHF patients compared with healthy individuals, but decreased by 10 times in patients with DHF compared to the CHF group or the healthy group (Figure 6C). Interestingly, the participants with DHF also had a lower concentration of EV-Hsp47 than the participants with CHF or healthy participants (Figure 6D). Given these results, the sEVs, EV-hERG1, and EV-Hsp47 can be used to further characterize CHF and DHF pathophysiologies. To this end, we studied the correlation between (i) total sEVs and EV-hERG1 and (ii) total sEVs and EV-Hsp47 for both compensated as well as decompensated heart failure participants (Figure 7). The results of this analysis established that the levels of EV-hERG1 and EV-Hsp47 in relation to the total sEVs decreased in the decompensated heart failure group compared with the compensated heart failure participants, suggesting a potential means to differentiate these two types of patients.

In summary, hERG1 and Hsp47 are detectable in sEVs from cardiomyocytes and induced by hypoxia. In addition, in a clinical setting, the number of hERG1 and Hsp47 proteins in sEVs increased with the development of cardiac ischemia during the stress test. Moreover, the abundances of hERG1 and Hsp47 proteins in relation to the total sEVs decreased under heart failure conditions.

## 3. Discussion

Given the high prevalence of CVDs, which are the leading cause of death globally, the scientific and clinical communities are carrying out abundant research to develop better strategies for earlier diagnosis and management of CVD. Small extracellular vesicles are membrane-bound vesicles released by all cell types. Over the last decades, their components have been studied to identify potential prognostic and diagnostic molecules in a variety of diseases, including CVD [31]. Our data (i) provide the first evidence to conclude that the potassium channel hERG1 and the collagen chaperone Hsp47 are present in the membranes of sEVs from human peripheral blood; (ii) show that part of the sequences ^430^AFLLKETEEGPPATE^445^ for hERG1 and ^169^ALQSINEWAAQTTDGKLPEVTKDVERTD^196^ for Hsp47 are exposed to the extracellular domain of small vesicles from human blood plasma and a human cardiomyocyte cell line; (iii) in a pilot analysis indicate a positive correlation between myocardial ischemia, EV-hERG1, and EV-Hsp47; especially, sEVs between 60 and 120 nm for EV-hERG1 and 120 and 300 nm for EV-Hsp47 increased as a result of cardiac ischemia induced during the stress test; (iv) show another small group of participants with chronic heart diseases, that EV-hERG1 and EV-Hsp47, in relation to the total sEVs, were differentially expressed in compensated and decompensated heart failure cases. Currently, we are developing a more robust ELISA prototype to quantify the levels of EV-hERG1 and EV-Hsp47 in blood samples and validate our results.

### 3.1. hERG1 in Cardiovascular Diseases

The human ether-a-go-go-related gene (hERG) encodes the pore-forming subunit of a delayed rectifier voltage-gated K+ (VGK) channel [32]. The hERG is strongly expressed in the mammalian heart, and participates in rapidly activating the delayed rectifier K+ current (I_Kr_). Inherited mutations in this gene that lead to dysfunction in the hERG, cause long QT (LQT2) syndrome and sudden death, which occur in patients with cardiac ischemia [33]. The three described members of the family channels are the Kv11.1 (hERG1), Kv11.2 (hERG2), and Kv11.3 (hERG3). The hERG1 contains six transmembrane segments (S1–S6), with S1–S4 contributing to the voltage sensor domain (VSD), and S5–S6, along with the intervening pore loop, contributing to the pore domain [32]. 

In addition, hERG1 channels have also been found to be involved in modulating cell proliferation and apoptosis [34]. Therefore, the hERG1 plays a protective role in sepsis-induced cardiac dysfunction (SICD) via regulation of FAK/AKT-FOXO3A to block LPS-induced myocardium apoptosis, indicating a potential effect of the potassium channels in the pathophysiology of SICD.

hERG1 function is insensitive to PKA or PKC phosphorylation modulation per se, but can be impaired by the activators of PKA or PKC with prolonged exposure via the generation of ROS. It has been proposed that hERG impairment due to ROS accumulation (induced by PKA and PKC) contributes to impaired cardiac repolarization, potentially contributing to many pathological conditions of the heart, such as heart failure [35,36].

In vitro studies showed that overexpression of the hERG in human embryonic kidney (HEK) cells and neonatal rat cardiomyocytes exposed to hypoxia experienced a reduction in hERG activity and expression [37]. In addition, the hERG/I_Kr_ channel was selectively cleaved by serine proteases, such as proteinase K (PK) in the S5-pore linker of the hERG, and calpain-1, which is upregulated in cardiac ischemia, and in the Gly-603 in the S5-pore linker of the hERG [38]. This evidence provides new insight into the relationship between hypoxia and ischemic heart disease, suggesting that damage to the hERG mediated by proteases may contribute to ischemia-associated QT prolongation and sudden cardiac death. Our hypothesis is that cleaved hERG proteins due to hypoxia could be eliminated from the cell by sEVs, and this could explain why we found increased levels of hERG1 in sEVs with hypoxia.

In another study on heart failure (HF) patients with dilated cardiomyopathy (DCM) vs. ischemic cardiomyopathy (ICM) as end stages, there was no difference in hERG1 mRNA levels within the cardiac tissue [39]. However, it has been described that the I_Kr_ blockade is reduced in HF, suggesting decreased functional hERG1 expression. This attenuated functional response is associated with altered hERG1a:hERG1b protein stoichiometry in the failing human left ventricle. Thus, I_Kr_ protein and functional expression may be important determinants of repolarization remodeling in failing left ventricles in humans [40].

In this study, we report for the first time the presence of hERG1 in the membrane of extracellular vesicles in peripheral blood samples, using an anti-hERG1 antibody that recognizes the region between the S1 and S2 domains of the hERG1, suggesting that the presence of hERG1 on the surface of the extracellular vesicles may reflect damaged conditions of the cardiac tissue.

### 3.2. Hsp47 in Cardiovascular Diseases

Hsp47, which is encoded by the SERPINH1 gene, is a member of the serpin superfamily of serine proteinase inhibitors. This protein is localized to the endoplasmic reticulum, and plays a role in collagen biosynthesis as a collagen-specific molecular chaperone. Hsp47 is a molecular chaperone that directly interacts with pro-collagens, and is associated with many fibrotic diseases [41].

Hsp47 in myofibroblasts is the primary mediator of tissue fibrosis and scar formation in the injured adult heart, which unexpectedly affects cardiomyocyte hypertrophy. Hsp47 of cardiac fibroblasts aggravates cardiac fibrosis postmyocardial ischemia and reperfusion injury [42].

Recently, single-cell transcriptome analysis of noncardiomyocytes showed that Hsp47 was associated with obese cardiac fibrosis [43]. Moreover, data from microarrays showed that Hsp47 is expressed only during the acute phase of a stroke [44]. Furthermore, the level of Hsp47 significantly increased with aging and obesity in mouse cardiac endothelial cells [45]. The undoubted role of this protein in modulating cardiac fibrosis and its relationship with myocardial infarction makes it a potential biomarker and/or therapeutic target for myocardial conditions.

### 3.3. hERG1 and Hsp47 on the Surface of sEVs

Our research provides the first evidence that hERG1 and Hsp47 are present on the surfaces of sEVs in peripheral blood. In consequence, it is possible to quantify sEVs with these two proteins and study their association with cardiovascular diseases. Currently, altered HERG1a/1b stoichiometry in human heart failure has been documented [46]. However, neither hERG1 nor Hsp47 have been detected on the surface of extracellular vesicles in blood and associated with a clinical cardiac event.

Hsp47 is an intracellular chaperone protein that is vital for collagen biosynthesis. In addition, Hsp47 is present on the surfaces of platelets, having an extracellular role, where it interacts with collagen, stabilizing platelet adhesion and thrombus formation. The ability of chaperone proteins to function in the extracellular environment may be important in a number of pathophysiological circumstances. It is unclear how Hsp47 exerts this effect, whether through modulation of ligand structure, or if it functions as an adhesion protein to enhance platelet–collagen interactions [47]. Using a commercial antibody against the “TKDVERTDGAL” sequence of human Hsp47, we found a positive signal for Hsp47 via flow cytometry (Figure 1 and Figure 5) and NTA (Figure 3 and Figure 5) in sEVs from human platelet-free plasma. In addition, we observed Hsp47 on sEV surfaces from the human cardiomyocyte cell line (Figure 2). This evidence suggests that Hsp47 binds in unknown ways to the surfaces of sEVs (Figure 8).

The sources of sEVs in the blood should be from different kinds of cells, with the main contribution being from hematopoietic cells, and with only a few derived from a different tissue. In this study, we provide evidence that the human cardiomyocyte cell line releases these two proteins on the surfaces of sEVs under “in vitro” normoxia conditions (21% O_2_), and is induced after 1 h of exposure to hypoxia (1% O_2_). So far, we do not understand the physiological or physiopathological reasons for the secretion of EV-hERG1 and EV-Hsp47. We are conducting experiments to understand the role of EV-hERG1 and EV-Hsp47 in the mechanisms of cardiovascular diseases, and their potential as biomarkers for this kind of disease. The in vitro data from cardiomyocyte cell culture suggests that the cell line releases EV-hERG1 and EV-Hsp47 to the extracellular media, and hypoxia increases these levels. However, considering our data, we cannot say that the EV-hERG1 and EV-Hsp47 found in blood samples come only from heart cardiomyocytes exposed to cardiac ischemia.

The results are very interesting regarding the two types of heart disease. In the stress test participants, we found that when transient cardiac ischemia is provoked, blood levels of EV-hERG1 and EV-Hsp47 significantly increased after 15 min of finishing the test. On the other hand, in the heart failure participants, the levels of total sEVs and the sEVs containing hERG1 and Hsp47 decreased in the samples from participants with decompensated heart failure, compared with compensated heart failure and healthy individuals. Moreover, the relationship between the levels of EV-hERG1 and total sEVs decreased in the decompensated heart failure group compared with the compensated group. Similar results were observed for the levels of EV-Hsp47. The stress test induces an acute cardiac condition, while heart failure is a chronic condition. Therefore, we considered it appropriate to only compare within each group because of the evident clinical differences between both groups. In this way, as mentioned above, we are preparing an ELISA kit to carry out a clinical study with a robust number of samples, in order to be able to compare different cardiovascular conditions. Thus, in the present state of the research, it is not recommended to compare both kinds of cardiovascular conditions.

New potential biomarkers could contribute to the diagnosis and eventual treatment of cardiovascular diseases (CVDs), in relation to numerous mutually corresponding factors involved in the pathogenesis of different CVD phenotypes [48]. Currently, circulating troponin I (cTn) in blood is the gold standard biomarker to determine myocardial tissue necrosis; therefore, it is a poor biomarker for early events related to myocardial infarction [49]. On the other hand, the natriuretic peptide is the gold standard biomarker for heart failure. However, its broad range of expression in other pathologies, such as infections, brain trauma, and adipose tissue dysfunction, makes it nonspecific in some situations [50]. The present results provide new evidence suggesting that hERG1 and Hsp47 in sEVs can be potential biomarkers of CVD.

## 4. Materials and Methods

### 4.1. Human Participants

The Ethics–Scientific Committee of Clínica Dávila, Santiago, Chile, approved the human study. The Institutional Review Board (IRB) of Clínica Dávila approved the study’s blood sampling protocol and methods (approval no. 27 May 2022). In addition, all procedures followed in this investigation were performed according to the principles outlined in the Declaration of Helsinki. 

#### 4.1.1. Stress Test Participants

We enrolled 120 patients in the study, and 13 received a positive diagnosis of cardiac ischemia in the stress test conducted by the cardiologists from Clínica Dávila. In order to have the same number of participants in both groups, 13 patients were randomly included with a negative diagnosis of cardiac ischemia in the stress test. The inclusion criterion for the stress test was an age ≥ 18 years at the time of signing the informed consent. Blood samples were taken from the participants, whether positive or negative for ischemia, 15 min after the cardiac stress test. The determination of a positive or negative diagnosis of cardiac ischemia was conducted by a team of cardiologists headed by Dr. Ricardo Larrea, from the Cardiology Department of the Clínica Dávila, and a positive diagnosis was confirmed by coronary angiogram.

#### 4.1.2. Participants with Chronic Heart Failure Disease

The cardiology team at the Dávila Clinic invited patients with heart failure who are regularly treated at this institution to participate in this study. Participants were selected at random until there were 10 participants in each group for the purposes of this research, according to the following inclusion and exclusion criteria [51]:

##### Decompensated Heart Failure (DHF) Inclusion Criteria

Symptoms and signs characteristic of the clinical condition (Framingham criteria).Elevated levels of NT-ProBNP above 1200 pg/mL (normal value < 450 pg/mL).An echocardiogram consistent with either of the two following situations:
A left ventricular ejection fraction (LVEF) less than 50% (i.e., LVEF depressed).An LVEF greater than or equal to 50%, if accompanied by structural changes in the heart (i.e., LVEF preserved).

##### Compensated Heart Failure (CHF) Inclusion Criteria

A known history of heart failure treatment.A concordant echocardiogram (according to the same criteria described for the DHF echocardiogram).A functional capacity of I or II, according to the New York Classification (N.Y.H.A.).Compensated for at least three months without variation in symptoms or functional capacity.Administration of at least three out of the four drugs recommended by the clinical guidelines at effective doses (ruling out the initial stage of dose escalation): (i) sacubitril/valsartan, angiotensin-converting enzyme (ACE) inhibitors, and angiotensin II receptor blockers; (ii) beta blockers; (iii) spironolactone; and (iv) sodium–glucose cotransporter 2 (SGLT2) inhibitors.

##### Heart Failure Exclusion Criteria

The presence of an active concomitant inflammatory or infectious disease.Undergoing systemic immunosuppressive and/or steroidal therapy.

### 4.2. Sample Storage

Blood samples were collected (10–35 mL) using ethylenediaminetetraacetic acid (EDTA). Fresh blood was centrifuged for 15 min at 1500× *g* and 4 °C to obtain platelet-free plasma (PFP). The PFP was transferred to cryotubes and stored at −80 °C [17].

### 4.3. Cell Culture

(A) Primary cardiomyocyte culture: Rats were bred at the Animal Breeding Facility at the Facultad de Ciencias Químicas y Farmacéuticas, Universidad de Chile (Santiago, Chile). All of the experiments were conducted with the approval of the Universidad de Chile Institutional Bioethical Committee, and in accordance with the National Institutes of Health (NIH) Guide for the Care and Use of Laboratory Animals, as well as the guidelines in Directive 2010/63/EU of the European Parliament on the protection of animals used for scientific purposes. For each cardiomyocyte culture, a total of 10 neonatal rats were used. Cardiomyocytes were isolated enzymatically from neonatal Sprague–Dawley rats (P0-1), as previously described [52,53]. In brief, newborn rats were placed on a cold surface (20 min) and administered light anesthesia. The animals were euthanized by decapitation, and the hearts were collected; the ventricles were separated from the atria and cut into small pieces in a saline solution containing (mM) 0.8 MgSO_4_ (7H_2_O), 116 NaCl, 5.4 KCl, 0.8 NaH_2_PO_4_ (4H_2_O), 5.6 glucose, and 20 HEPES (pH 7.4) to obtain a homogeneous suspension. Myocardial cells were dissociated by enzymatic digestion using 3 U/mL papain and 0.2 mg/mL type II collagenase (Invitrogen, Carlsbad, CA, USA) in a 100 μM CaCl_2_ saline solution. Tissue pieces were incubated for 15 min at 37 °C under constant agitation. Thereafter, the supernatant was removed, and the cells were washed with a saline solution (without CaCl_2_) three to five times. Then, the cells were incubated (15 min at 37 °C, by agitation) in a saline solution containing Ca^2+^ (1.2 mM), pancreatin (1.6 mg/mL, Sigma–Aldrich, St. Louis, MO, USA), and type II collagenase (0.2 mg/mL, Invitrogen, Carlsbad, CA, USA). The supernatant was then withdrawn and centrifuged, and the cells were resuspended in DMEM (Invitrogen, Carlsbad, CA, USA) supplemented with 10% horse serum, 5% fetal bovine serum (FBS), and 1% antibiotic (penicillin/streptomycin). This last step was repeated five times. Once collected, the cells were seeded onto a culture plate and incubated for 3–4 h in an incubator (5% CO_2_) at 37 °C to allow for fibroblast adhesion. Subsequently, the supernatant was collected and centrifuged for 5 min at 900× *g*. The collected cells were resuspended in Dulbecco’s Modified Eagle’s Medium (DMEM) and seeded onto plates pretreated with gelatin (1%). Sixteen hours later, the medium was replaced with DMEM plus 0.5% FBS [54]. (B) Cardiomyocyte cell line: the cell line (AC-16, Merck) was cultured in Dulbecco’s Modified Eagle’s Medium (DMEM)/F12 (Sigma–Aldrich, St. Louis, MO, USA, Cat. D6434) and supplemented with 2 mM L-glutamine (Corning, Darmstadt, Germany, Cat. 25-005-CI), 12,5% FBS (Gibco, Life Technologies, Grand Island, NY, USA, Cat. 16140071), and penicillin–streptomycin (Gibco, Life Technologies, Grand Island, NY, USA, Cat. 15140122) [54]. (C) HEK293 cells: the HEK293 cell line was cultured in Dulbecco’s Modified Eagle’s Medium (DMEM) (Sigma–Aldrich, St. Louis, MO, USA) and supplemented with 2 mM L-glutamine (Corning, Darmstadt, Germany, Cat. 25-005-CI), 10% fetal bovine serum (FBS) (Gibco, Life Technologies, Grand Island, NY, USA, Cat. 16140071), and penicillin–streptomycin (Gibco, Life Technologies, Grand Island, NY, USA, Cat. 15140122) [55].

### 4.4. Hypoxia Cell Culture

Cells were cultured in DMEM plus 0.5% FBS (free of extracellular vesicles from the FBS). Hypoxia was set to 1% of O_2_ using gas displacement (95% N_2_ and 5% CO_2_). To establish the duration for exposure to hypoxia, the cultures were subjected for different durations (15 and 30 min and 1, 2, 4, and 8 h) [55].

### 4.5. NTA Measurements with the NanoSight NS300

All samples were diluted in Dulbecco’s phosphate-buffered saline (DPBS, Sigma–Aldrich, St. Louis, MO, USA) to a final volume of 1.5 mL. The starting dilution was 1:250, and the measurement concentrations were determined by pretesting the ideal particles per frame value (20–100 particles/frame); if the concentration was outside the range, the sample was diluted by 2 orders of magnitude and tested again until the reading was within the set range of particles per frame. All of the counts were performed in five replicates for each sample group. The data used in our analysis were collected from 30-second videos with the detection threshold (DT) set at 5. The experiment’s videos were analyzed using NTA 3.2 software (Malvern Panalytical Ltd., Malvern, UK). Washes were carried out with DPBS before the introduction of the first sample, between samples, and at the end of the reading, in order to avoid any residues in the system [17].

### 4.6. NTA Fluorescence

The PFP samples were analyzed to measure the sEV concentrations by NTA. An aliquot of FPF was used to prepare 1.5 mL of 1 × 10^9^ particles/mL solution in DPBS. The diluted samples were incubated for 1 h at 37 °C with either rabbit anti-hERG1 (Alomone labs, Jerusalem, Israel, #APC-109) or rabbit anti-HSP47 (MyBioSource, Inc., San Diego, CA, USA, #MBS9208399), which recognizes the ^430^AFLLKETEEGPPATE^445^ or ^169^ALQSINEWAAQTTDGKLPEVTKDVERTD^196^ sequences, respectively. Then, the Alexa Fluor 532 (Invitrogen, Carlsbad, CA, USA) antibody was incubated for one additional hour at 37 °C to detect the anti-hERG1 or anti-Hsp47 antibodies. The samples were analyzed by fluorescence with NTA. The NanoSight was set to camera level 9 for the samples analyzed in the light scatter mode (LSM), and from 15 to 16 for the samples analyzed in the fluorescence mode (FM). The DT was set to 5 for all samples for both the LSM and FM [20].

### 4.7. sEV Purification by Ultracentrifugation

Thawed PFP samples were centrifuged at 12,000× *g* for 45 min at 4 °C. The supernatant was centrifuged at 110,000× *g* for 120 min at 4 °C. The supernatant was discarded, and the pellet was resuspended in 1× PBS, and then centrifuged at 110,000× *g* for 60 min at 4 °C. Finally, each pellet was resuspended in 200 μL of 1× PBS and stored at −80 °C until used [17].

### 4.8. sEV Purification by Size Exclusion Chromatography

The PFP was thawed from its storage at −80 °C. The qEV columns (Izon Science, Christchurch, New Zealand) were used in the size exclusion chromatography (SEC) to separate the contaminating soluble protein from the sEVs according to the manufacturer’s instructions. The qEV columns removed approximately 99% of the soluble protein, delivering highly purified samples of sEVs. Following the isolation of the sEVs, they were studied using cytometry analysis [56].

### 4.9. Western Blot Analysis

SDS-PAGE was performed using DTT (Calbiochem, San Diego, CA, USA) in sEVs obtained by ultracentrifugation from the PFP. Approximately 25 μg of the total protein of each sEV sample was mixed with 4× Laemmli Sample Buffer, heated (95 °C, 5 min), and loaded into a 12% polyacrylamide gel. The gel was run at 80 V until the dye front crossed the junction between the concentrator and separator gels (approximately 30 min). The voltage was then increased to 120 V, and the total duration of the electrophoresis was 2 h. The proteins were transferred to a nitrocellulose membrane (Thermo Scientific, Carlsbad, CA, USA) and blocked (Odyssey blocking buffer, Li-cor). The primary antibody was either anti-hERG1 (Alomone labs, Jerusalem, Israel, # APC-109; 1:1000) and anti-Hsp47 (MyBioSource, Inc., San Diego, CA, USA, # MBS 9208399; 1:1000). The negative sEVs markers [57] were studied using anti-calnexin (Cell Signaling, Danvers, MA, USA, #2679), anti-Grp94 (Cell Signaling, Danvers, MA, USA, #2104), or the positive anti-CD9 (ABCAM, Cambridge, CB2 0AX, UK, # Ab65230; 1:500) sEVs marker. The secondary antibody was tagged with infrared dye anti-rabbit 680 (A21076) or anti-rabbit 750 (A21039) Alexa Fluor-conjugated antibody (Life Technologies, Carlsbad, CA, USA; 1:10,000). Membranes were scanned using an Odyssey CLx (Li-cor, Lincoln, Nebraska USA). The incubation period for each primary antibody was overnight (16 h) in a rocker-type shaker at 70 rpm and 4 °C. The secondary antibodies were incubated for 1 h at room temperature under the same agitation conditions [55].

### 4.10. Transmission Electron Microscopy (TEM)

Briefly, the pool of sEVs purified by ultracentrifugation were fixed for 4 h (4% formaldehyde in PBS 0.1 M pH 7). The samples were washed overnight with double distilled water to remove the excess of 4% paraformaldehyde. The samples were incubated with 1% aqueous osmium tetroxide for 45 min, and then washed 3 times with bidistilled water for 10 min. Subsequently, the samples were incubated in 0.5% aqueous uranyl acetate for 1 h, and 10 min washes were performed in ethanol at different dilutions (50%, 70%, 95%, and 100%). Incubation in LR White:ethanol resin (1:1) was performed overnight, and the samples were left embedded in resin for a minimum of 4 h for the ethanol to evaporate. Finally, the samples were embedded in gelatin capsules and filled with LR White acrylic resin, polymerized with UV light for 30 min, and then ultrathin cuts of 90 nm thickness were made and collected with nickel grids. Then, the samples were washed with 0.02 M Tris buffer at pH 7.4. The grids were exposed to primary antibodies that recognized hERG1 or Hsp47, and then to species-specific anti-IgG antibodies conjugated to colloidal gold particles (6 nm or 12 nm) (Jackson ImmunoResearch Laboratories, West Grove, PA, USA). After washing, the membranes underwent negative staining with 0.5% uranyl acetate for 20 min. Finally, the samples were examined using a transmission electron microscope (Hitachi HT7700; 120 kV) at the Microscopy Facility Universidad de Santiago de Chile (USACH; Chile) [58].

### 4.11. Flow Cytometry

The evaluation of hERG1 and Hsp47 in the isolated sEVs was performed as we previously described, with some modifications [59]. Briefly, 3.5 × 10^8^ particles were incubated with aldehyde/sulfate latex beads (Molecular Probes, United States, Cat. A37304) overnight at 4 °C. Subsequently, 1 M glycine (United States Biological, Salem, MA, USA, Cat. G8160) was added to the samples for 1h at RT and centrifuged at 8000× *g* for 2 min at 4 °C. The pellets were resuspended in 10% *w*/*v* bovine serum albumin (BSA; Winkler Ltd.a., Santiago, Chile, Cat. BM-0150) for 45 min at RT. Then, the samples were mixed with a 2% BSA solution containing anti-hERG1 (Alomone labs, Jerusalem, Israel, #APC-109), anti-Hsp47 (MyBioSource, Inc., San Diego, CA, USA, #MBS9208399) as the primary antibodies, or rabbit IgG (Alomone labs, Jerusalem, Israel, #RIC-001). In addition, anti-CD9 (BD Pharmingen, San Diego, CA, USA, Cat. 555370), anti-CD63 (BD Pharmingen, San Diego, CA, USA, Cat. 556019), anti-CD81 (BD Pharmingen, San Diego, CA, USA, Cat. 555675), or mouse IgG1 (BD Biosciences, San Jose, CA, USA, Cat. 349040) for 30 min at RT was also tested. The immunolabeled particle-coupled beads were centrifugated at 8000× *g* for 2 min at 4 °C. The pellets were washed, incubated with 10% BSA solution for 30 min at RT, centrifugated, and washed again with 1× PBS. The pellets were resuspended in a solution containing 2% BSA and secondary antibody α-mouse IgG1 Alexa Fluor 488 (Biolegend, San Diego, CA, USA, Cat. 406626) or secondary antibody α-rabbit IgG Alexa Fluor 647 (Biolegend, San Diego, CA, USA, Cat. 406414) for 30 min at RT. Finally, the sample was washed several times, and the pellet was resuspended in 1x PBS for acquisition in the CantoTM II cytometer (BD Biosciences, San Jose, CA, USA). The data acquired were analyzed using FlowJo software V10 (Tree Star, Ashland, OR, USA) [60].

### 4.12. In-House ELISA

All of the reagents and samples were warmed to room temperature (18–25 °C) before use. The appropriate wells were covered overnight at 4 °C with 100 µL of 1 ng/µL anti-hERG1 or anti-Hsp47 antibody solution at 120 rmp under rocking conditions. The next day, the solution was discarded, and each well was washed 4 times with DPBS using a multichannel pipette. Then, each well was incubated for 1 h with a commercial blocking solution (StartingBlock^TM^ (PBS) Blocking buffer, Thermo Scientific, Carlsbad, CA, USA). The blocking solution was discarded, and each well was washed 4 times with DPBS using a multichannel pipette. After the last wash, the remaining DPBS was removed by decantation, inverting the plate and blotting it against clean paper towels. Then, 100 µL of DPBS, or a volume of culture media equivalent to 1 × 10^9^ sEVs diluted in DPBS from cardiomyocytes exposed to normoxia and hypoxia, was incubated for one hour at room temperature and 120 rpm in a rocking system. The analyte solution was discarded, and each well was washed four times with DPBS using a multichannel pipette. Then, 100 µL of 1 ng/µL anti-CD63 (Novusbio, Centennial, CO, USA) or anti-CD81 (SinoBiological (Beijing, China) antibody solution conjugated with horseradish peroxidase (HRP) was incubated for one hour at room temperature at 120 rpm in the rocking system. The solution was discarded, and each well was washed 4 times with DPBS using a multichannel pipette. A volume of 100 µL of 3,3′,5,5′-Tetramethylbenzidine (TMB) Reagent (Thermo Scientific, Carlsbad, CA, USA) was added to each well and incubated for 30 min at room temperature in the dark with gentle shaking. Finally, 50 µL of Stop Solution (Thermo Scientific, Carlsbad, CA, USA) was added to each well. Immediately, the signal was quantified using a TCAN spectrophotometer (Tecan Group Ltd., Männedorf, Switzerland) for its absorbance at 450 nm [30].

### 4.13. Statistical Analysis

The data are expressed in terms of the mean ± SD. A statistical comparison was conducted with GraphPad software (Prism 6 Mac OS X) using one-way ANOVA and Wilcoxon–Mann–Whitney tests. Significance was set at *p* < 0.05.

## 5. Conclusions

Taken together, our findings reveal that small extracellular vesicles secreted by cultured human cardiomyocytes, as well as those present in circulating blood, express hERG1 and Hsp47 proteins on their surfaces. When the cardiomyocytes were exposed to hypoxia or when cardiac ischemia was induced in humans, the secreted and circulating sEV levels increased. In addition, the hERG1 and Hsp47 proteins in the sEVs were differentially abundant in the heart failure cases. Based in our preliminary data, we propose sVE-hERG1 and sVE-Hsp47 in peripheral blood as biomarkers of cardiovascular diseases (Figure 8).

## 6. Patents

A patent application entitled ‘Methodology for In Vitro Diagnosis (IVD) of Cardiovascular Disease’ was filed in the National Institute of Industrial Property (INAPI)-Chile.

## Figures and Tables

**Figure 1 ijms-25-04913-f001:**
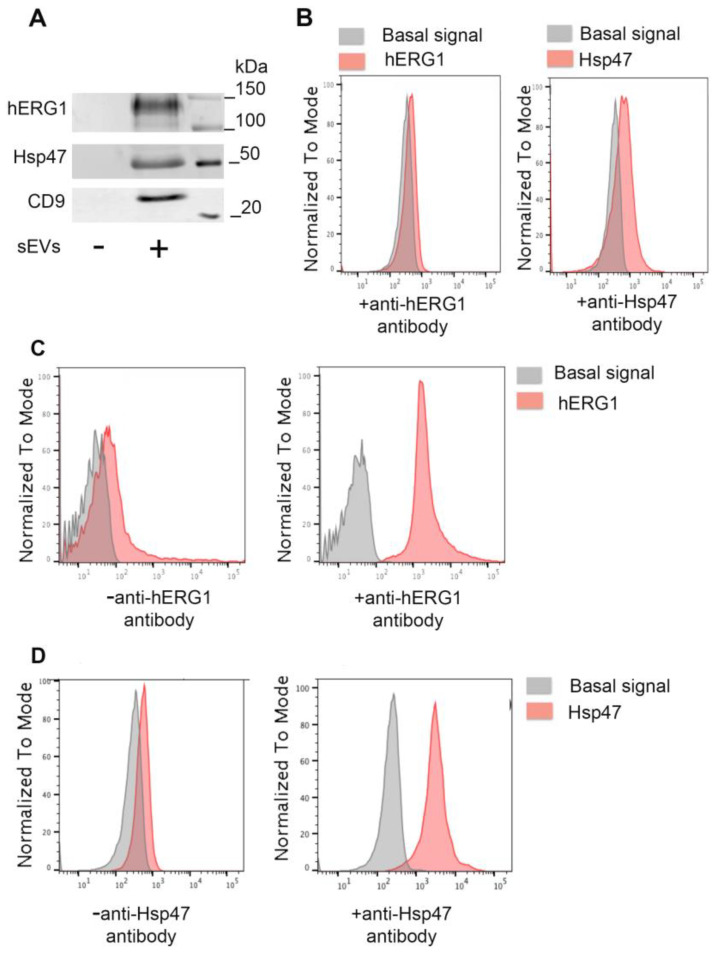
The hERG1 (EV-hERG1) and Hsp47 (EV-Hsp47) epitopes are expressed on the surfaces of sEVs from cardiac ischemia patients. (**A**). Pooled sEV samples from individuals with cardiac ischemia were purified from platelet-free plasma by several ultracentrifugation steps and tested via Western blotting using anti-hERG1, anti-Hsp47, and anti-CD9 antibodies. Both the sEV-free fraction (−, “12,000× *g* pellet”) and the sEV-enriched fraction (+, “110,000× *g* pellet”) were analyzed. (**B**). Flow cytometry: PLP fraction produced by centrifugation at 12,000× *g* was incubated with anti-hERG1/α-rabbit IgG Alexa Fluor 647 or anti-Hsp47/α-rabbit IgG Alexa Fluor 647 antibodies. (**C**). Flow cytometry: purified sEVs from blood plasma enriched by ultracentrifugation were incubated with or without anti-hERG1, then incubated with α-rabbit IgG Alexa Fluor 647 antibodies. (**D**). Flow cytometry: purified sEVs from blood plasma enriched by ultracentrifugation were incubated with or without anti-Hsp47, then incubated with α-rabbit IgG Alexa Fluor 647 antibodies.

**Figure 2 ijms-25-04913-f002:**
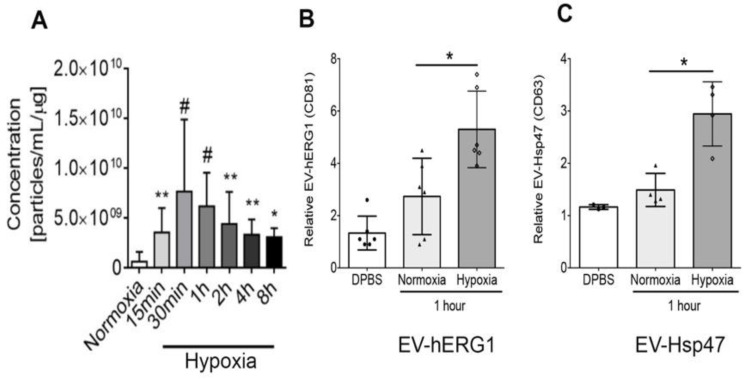
Extracellular vesicles containing hERG1 (EV-hERG1) and Hsp47 (EV-Hsp47) are secreted by cardiomyocytes at low concentrations of oxygen. (**A**) The sEV concentration was studied in primary cell culture from rat neonatal cardiomyocytes exposed to normoxia (21% oxygen) and hypoxia (1% oxygen) via nanoparticle tracking analysis (NTA). The bar graph represents the mean ± SD (n = 7 per group), and the statistical analysis was conducted using an ANOVA, Tukey’s post hoc test, and the Mann–Whitney Test. Significance is denoted as * *p* < 0.05 and ** *p* < 0.005 vs. normoxia, and ^#^
*p* < 0.05 vs. 15 min of exposure to hypoxia. (**B**) The EV-hERG1 concentration was studied in a human cardiomyocyte cell line (AC16) exposed to normoxia (21% oxygen) and hypoxia (1% oxygen) in an FBS-free culture medium for 1 h using in-house ELISA. The bar graph represents the mean ± SD (n = 6 per group), and the statistical analysis was conducted using the *t*-test and Mann–Whitney Test. Significance is denoted as * *p* < 0.05 vs. normoxia. (**C**) The EV-Hsp47 concentration was studied in a human cardiomyocyte cell line (AC16) exposed to normoxia (21% oxygen) and hypoxia (1% oxygen) in an FBS-free culture medium for 1 h using in-house ELISA. The bar graph represents the mean ± SD (n = 4 per group), and the statistical analysis was conducted using a *t*-test and Mann–Whitney test. Significance is denoted as * *p* < 0.05 vs. normoxia.

**Figure 3 ijms-25-04913-f003:**
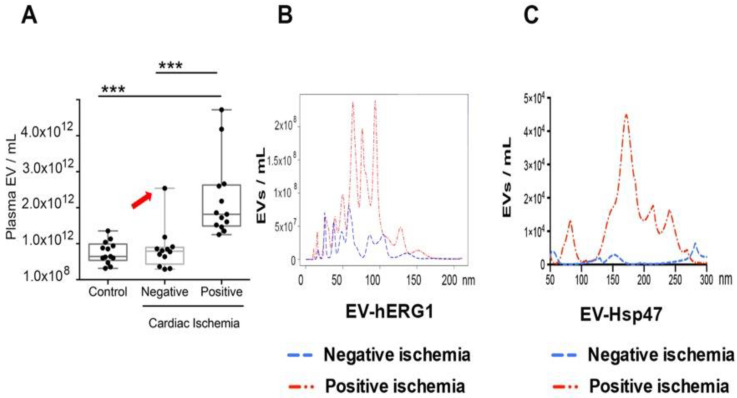
The levels of extracellular vesicles containing hERG1 (EV-hERG1) and Hsp47 (EV-Hsp47) increased in the blood samples during cardiac ischemia. The concentrations of sEVs containing hERG1 or Hsp47 were studied in platelet-free plasma (PFP) from blood samples of the stress test participants using nanoparticle tracking analysis. (**A**) The sEV concentration was studied in platelet-free plasma from the stress test participants using nanoparticle tracking analysis (NTA). The boxes represent the interquartile range of the values, whereas the whiskers’ spans represent the minima to maxima, showing all points for the control (n = 13), cardiac-ischemia-negative (n = 13), and cardiac-ischemia-positive (n = 13) samples. The red arrow point it out a participant with a negative cardiac ischemia diagnosis but with elevated levels of sEVs in their blood during the stress test. In B and C, the amounts of 1 × 10^9^ sEVs were determined for each blood plasma (PFP) sample and immunolabeled for 1 h with primary antibody, and then for 1 h with Alexa Fluor 532 secondary antibody. The graphic represents the averages for the patients with negative (n = 13) and positive (n = 13) cardiac ischemia diagnoses, as described in Table 1. The statistical analysis was conducted using ANOVA, Tukey’s post hoc test, and the Mann–Whitney Test. Significance is denoted as *** *p* < 0.0005. (**B**) anti-hERG1 antibody; (**C**) anti-Hsp47 antibody.

**Figure 4 ijms-25-04913-f004:**
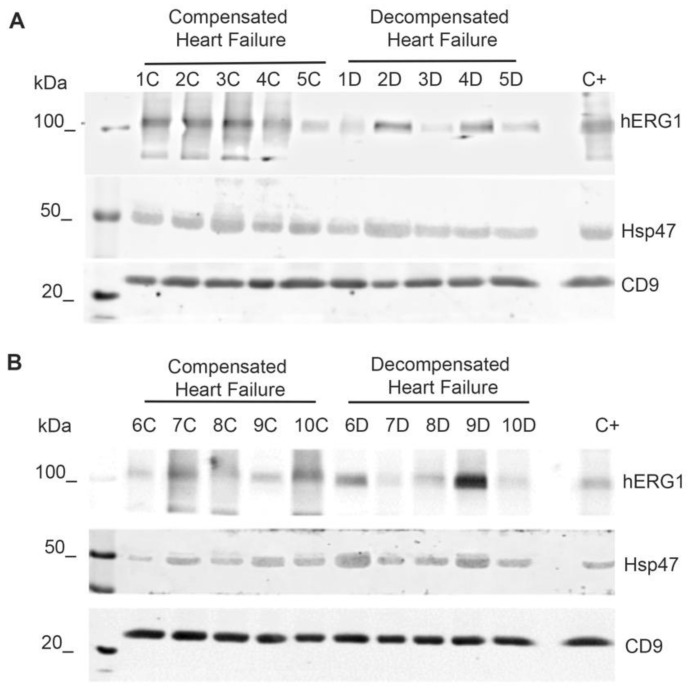
The expressions of hERG1 and Hsp47 were detected in extracellular vesicles from blood enriched by ultracentrifugation from decompensated heart failure and compensated heart failure participants. The sEVs were enriched by ultracentrifugation from 10 mL of platelet-free plasma from blood samples of 10 participants with compensated heart failure (CHF) (identified as 1C to 10C) and 10 participants with decompensated heart failure (DHF) (identified as 1D to 10D). The presence of the hERG1, Hsp47, and CD9 proteins (as markers of small extracellular vesicles) was determined via Western blot using specific antibodies for hERG1, Hsp47, and CD9, and 25 micrograms of total protein were taken from the sEVs of each patient. (**A**) Western blot analysis of 5 participants with CHF (1C–5C) and 5 participants with DHF (1D–5D). (**B**) Western blot of a second group of 5 participants with CHF (6C–10C) and 5 participants with DHF (6D–10D). C+ corresponds to a previously identified positive control of hERG1, Hsp47, and CD9 in purified sEVs.

**Figure 5 ijms-25-04913-f005:**
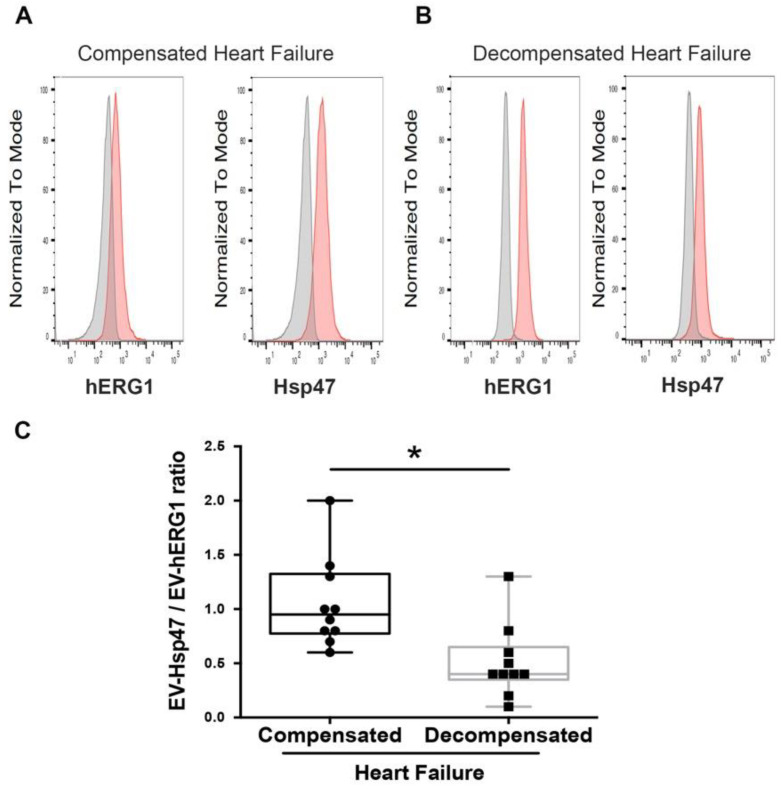
The expressions of hERG1 and Hsp47 were detected by flow cytometry in extracellular vesicles enriched by ultracentrifugation from compensated and decompensated heart failure participants. The sEVs were enriched by ultracentrifugation from 10 mL samples of platelet-free plasma (PFP) from 10 patients with compensated heart failure (CHF), and 10 with decompensated heart failure (DHF). The concentration of sEVs was determined using the nanoparticle tracking analysis (NTA) method for the NanoSight equipment. The presence of sEVs containing hERG1 (EV-hERG1) and sEVs containing Hsp47 (EV-Hsp47) was determined by flow cytometry, using a specific antibody for hERG1 and Hsp47 corresponding to 3.5 × 10^8^ purified sEVs. (**A**) Representative flow cytometry histograms of hERG1 and Hsp47 (basal signal in grey, hERG1 or Hsp47 signal in red) in the participants with CHF. (**B**) Representative flow cytometry histograms of hERG1 and Hsp47 (basal signal in grey, hERG1 or Hsp47 signal in red) in the participants with DHF. (**C**) The EV-Hsp47/EV-hERG1 ratio. The boxes represent the interquartile range of the values, whereas the whiskers’ spans represent the minima to maxima, showing all points for CHF (n = 10) and DHF (n = 10) participants. A statistical analysis was performed using ANOVA, Tukey’s post hoc test, and the Mann–Whitney test, with significance denoted as * *p* < 0.005.

**Figure 6 ijms-25-04913-f006:**
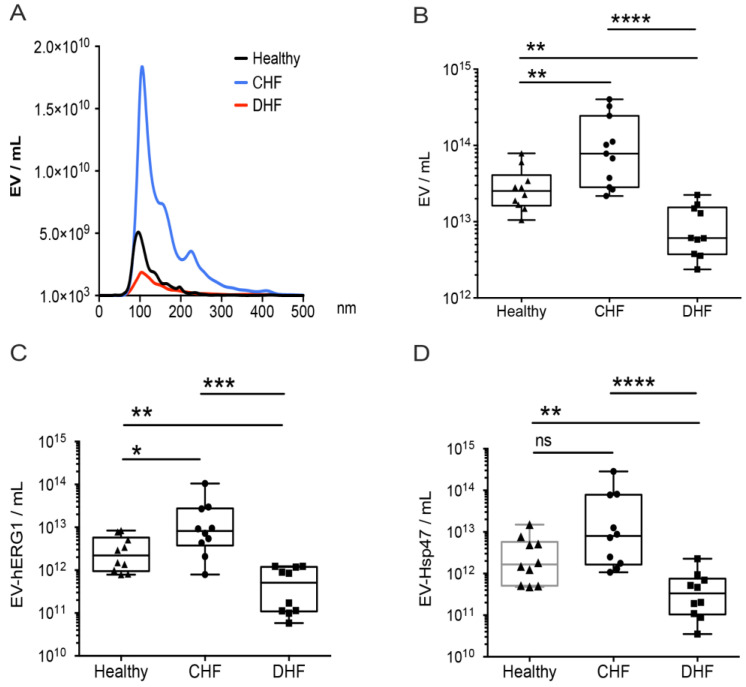
The presence of sEVs with hERG1 and Hsp47 decreased in the blood plasma of patients with decompensated heart failure. Platelet-free plasma from the blood of healthy (n = 10), compensated heart failure (CHF, n = 10), and decompensated heart failure (DHF, n = 10) participants was used to evaluate the concentrations of total sEVs, EV-hERG1, and EV-Hsp47 using the NTA method. (**A**) sEVs size distributions; (**B**) plasma sEV concentrations (EV/mL); (**C**) plasma EV-hERG1 concentrations (EV-hERG1/mL); (**D**) plasma EV-Hsp47 concentrations (EV-Hsp47/mL). The boxes represent the interquartile ranges of the values, whereas the whiskers’ spans represent the minima to maxima, showing all points for CHF (n = 10) and DHF (n = 10) participants. The statistical analysis was conducted using ANOVA, Tukey’s post hoc test, and the Mann–Whitney test. Significance is denoted as * *p* < 0.05; ** *p* < 0.005; *** *p* < 0.005, and **** *p* < 0.00005. ns, not significant.

**Figure 7 ijms-25-04913-f007:**
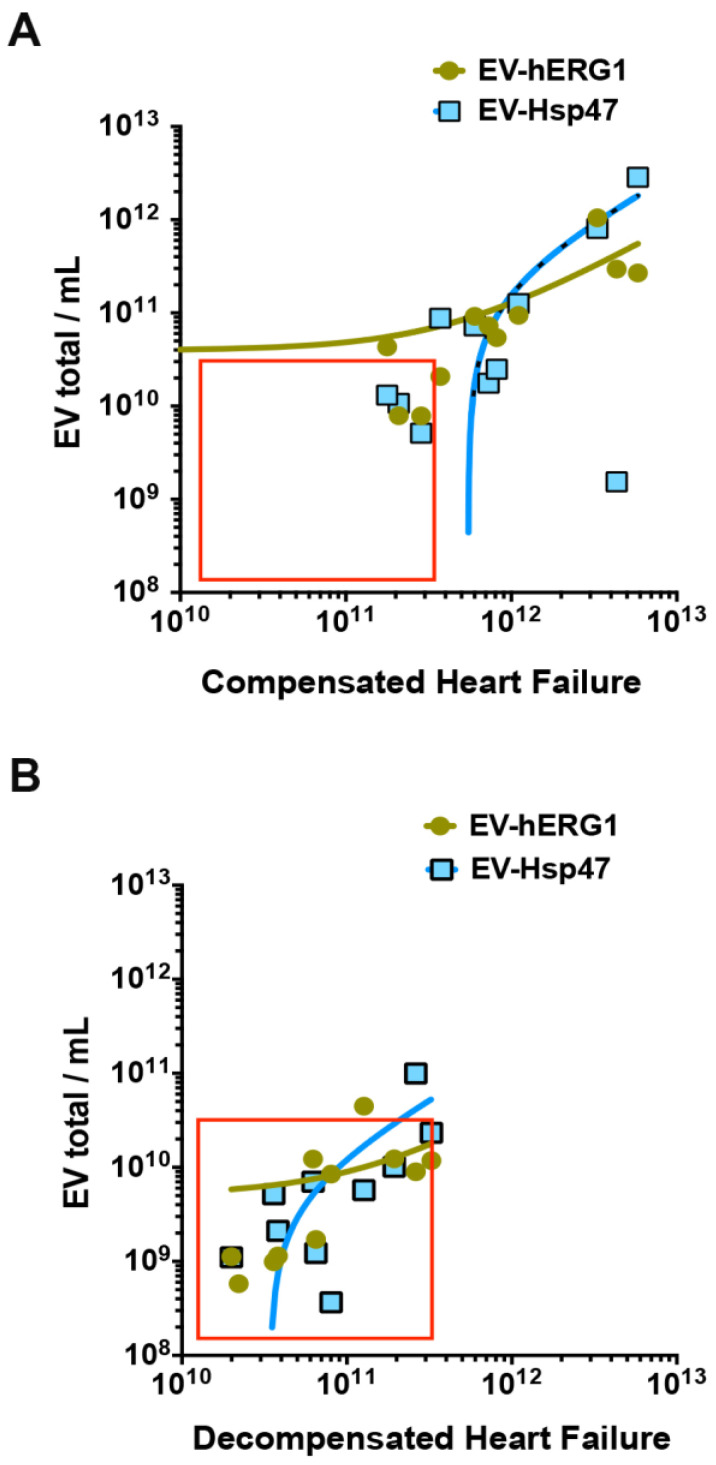
The concentrations of sEVs, EV-hERG1, and EV-Hsp47 as an means to discriminate between compensated and decompensated heart failure: (**A**) sEV (EV/mL) and EV-hERG1 correlation curve and the sEV (EV/mL) and EV-Hsp47 correlation curve for compensated heart failure; (**B**) sEV (EV/mL) and EV-hERG1 correlation curve and the sEV (EV/mL) and EV-Hsp47 correlation curve for decompensated heart failure. The red box represents participants with DHF.

**Figure 8 ijms-25-04913-f008:**
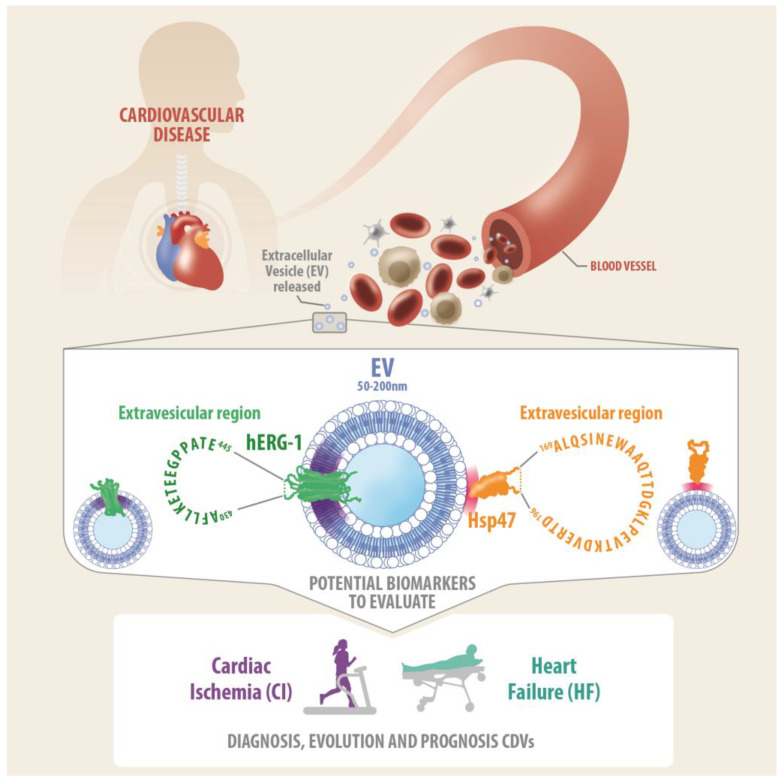
Potential biomarkers of cardiovascular diseases (CVDs). In blood samples, we found small extracellular vesicles (i.e., sEVs from 50 to 200 nm in size) that contained hERG1 and Hsp47 proteins on their surfaces. The results indicate that hERG1 and Hsp47 bind in unknown ways to the surfaces of sEVs. Our findings suggest that vesicular hERG1 and Hsp47 can be explored as possible biomarkers of CVDs.

**Table 1 ijms-25-04913-t001:** Stress test participants.

Stress Test			
	Negative for Myocardial Ischemia (n = 13)	Positive for Myocardial Ischemia (n = 13)	*p*-Value
Sex (male, %)	84.62	53.84	0.101
Age (year, range)	61 ± 12.86 (33–77)	64.92 ± 10.03 (55–84)	0.513
AHT (%)	46.15	69.23	0.122
T2DM (%)	25.00	30.00	0.583
Dyslipidemia (%)	46.15	40.00	0.552
Smoking (%)	41.66	40.00	0.639
AMI (%)	18.18	36.36	0.318

**Table 2 ijms-25-04913-t002:** Compensated heart failure (CHF) and decompensated heart failure (DHF) participants.

	CHF (n = 10)	DHF (n = 10)	*p*-Value
Sex (male, %)	72.73	72.73	1.000
Age (years; mean ± SD)	65.27 ± 8.49	63.00 ± 17.54	0.711
SBP (mmHg; mean ± SD)	119.78 ± 22.77	146.18 ± 32.14	0.047
DBP (mmHg; mean ± SD)	72.56 ± 7.42	86.55 ± 23.86	0.157
Heart rate (beats/min; mean ± SD)	76.71 ± 9.81	87.63 ± 25.14	0.385
HTA (%)	72.73	63.64	0.050
Dyslipidemia (%)	0	18.08	0.238
DM (%)	36.36	36.36	1.000
CKD (%)	0	0	1.000
Smoking (%)	9.09	45.45	0.074
Implantable cardioverter–defibrillator (ICD)	36.36	18.18	0.318
One-sided analysis			

## Data Availability

The raw data supporting the conclusions of this article will be made available by the authors on request.

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
