# Peer review of "Levels of Small Extracellular Vesicles Containing hERG-1 and Hsp47 as Potential Biomarkers for Cardiovascular Diseases"

_ijms, 2024, doi:10.3390/ijms25094913_

Round 1
Reviewer 1 Report
Comments and Suggestions for Authors
General comments:
The manuscript is well written. The authors investigated “The levels of small extracellular vesicles containing hERG-1 and Hsp47 are potential biomarkers for cardiovascular diseases”. The study is interesting and adds to the existing body of knowledge. This study aims to characterize the presence of hERG1 and Hsp47 associated with the membrane of extracellular vesicles in blood samples to suggest potential biomarkers of cardiovascular disease. The data and results of the current study are well presented and the findings are beneficial to the pharmaceutical industry. However, there are a few things that need clarification and revisions.
Details comments:
1. Page 1, Abstract: Please add the objective of the current study. Rewrite the abstract as an unstructured abstract.
2. Page 1, Background: Please replace the "Background" with "Introduction".
3. Please change all the citation formats in the whole manuscript to "[number]" not "(number)".
4. Page 3, Methods: Participants; Please add the human ethics approval number and year of approval.
5. Page 3-4: Please add the citations for protocols, particularly for inclusion criteria.
6. Page 4, Sample storage: Please add the volume of blood collected and the citation for these protocols.
7. Page 4, Cardiomyocyte culture: Please include the total of rats used in the current study. Citations for the animal's euthanized tissue processing (aorta) should be added.
8. Page 5, Please add the citations for NTA fluorescence and extracellular purification.
9. Page 5, Western blotting: The SDS-PAGE should be rewritten (Please add the voltage and time for electrophoresis). Please add the incubation time for each antibody and the catalog number for each antibody.
10. Page 5, Flow cytometry: Please remove the DOI number for reference and add the reference number for citation.
11. References: The reference format should be revised and please change the format according to MDPI reference format.
Comments on the Quality of English LanguageEnglish language are fine and minor English editing are required.
Author Response
Reviewer 1.
General comments:
The manuscript is well written. The authors investigated “The levels of small extracellular vesicles containing hERG-1 and Hsp47 are potential biomarkers for cardiovascular diseases”. The study is interesting and adds to the existing body of knowledge. This study aims to characterize the presence of hERG1 and Hsp47 associated with the membrane of extracellular vesicles in blood samples to suggest potential biomarkers of cardiovascular disease. The data and results of the current study are well presented and the findings are beneficial to the pharmaceutical industry. However, there are a few things that need clarification and revisions.
Answer. We appreciate all the relevant reviewer`s suggestions that will undoubtedly improve the quality of our article. It is important to note that from the editor's office gave us only 10 days to respond to the reviewer, so we did not have time to run new experiments.
We have elaborated a detailed response to each comment and criticism of the reviewer. Thus, we have incorporated the changes in the new version of the manuscript.
Details comments:
- Page 1, Abstract: Please add the objective of the current study. Rewrite the abstract as an unstructured abstract.
Answer. We appreciate this important suggestion that certainly strengthens our study. We have followed your suggestion and clarified this point in the abstract of the new version of the manuscript (Please, see page 1): “ “Therefore, this study characterized the presence of hERG1 and Hsp47 in the membrane of extracellular vesicles (sEVs) from blood samples and their association with CVD”.
- Page 1, Background: Please replace the "Background" with "Introduction.
Answer. Thank you for this comment. It was corrected in the new version of the manuscript (Please, see page 1).
- Please change all the citation formats in the whole manuscript to "[number]" not "(number)".
Answer. Thank you for this observation. It was corrected in the new version of the manuscript.
- Page 3, Methods: Participants; Please add the human ethics approval number and year of approval.
Answer. Thank you for this recommendation. It was corrected in the new version of the manuscript (Please, see page 4): ”(approval no. 27-05-2022)”.
- Page 3-4: Please add the citations for protocols, particularly for inclusion criteria.
Answer. Thank you for this note. The citations for protocols were added in the new version of the manuscript.(page 4) Reference [10].
- Page 4, Sample storage: Please add the volume of blood collected and the citation for these protocols.
Answer. Thank you for this comment. It was corrected in the method section of the new version of the manuscript as: “Blood samples were collected (10-35 mL)”. Please, see page 5.
- Page 4, Cardiomyocyte culture: Please include the total of rats used in the current study. Citations for the animal's euthanized tissue processing (aorta) should be added.
Answer. Thank you for this comment. In the new version of the manuscript was included the total amounts of rats used and the paragraph was modified as: “ For each cardiomyocyte culture, a total of 10 neonatal rats were used.” (Please, see page 4).
We appreciate this important suggestion about the animal's euthanized tissue processing (aorta). The protocol was described before (references 29), which were cited in the first version of the manuscript (please see page 5).
- Page 5, Please add the citations for NTA fluorescence and extracellular purification.
Answer. Thank you for pertinent comment. The new information was added in the new version of the manuscript as reference 19.
- Page 5, Western blotting: The SDS-PAGE should be rewritten (Please add the voltage and time for electrophoresis). Please add the incubation time for each antibody and the catalog number for each antibody.
Answer. We appreciate this important suggestion that certainly improves the quality of the manuscript. In the new version of the manuscript all the paragraph corresponding to SDS was rewritten: “2.9 Western blot analysis. SDS-PAGE was performed using DTT (Calbiochem) in sEVs obtained by ultracentrifugation from the PFP. Approximately 25 μg of the total protein of each sEV was mixed with 4x Laemmli Sample Buffer, heated (95ºC, 5 min), and loaded into a 12% polyacrylamide gel. The gel was run at 80 V until the migration front crossed the gel concentrator–gel separator junction (approximately 30 min). The voltage was then increased to 120 V, and the total duration of the electrophoresis was 2 hours. The proteins were transferred to a nitrocellulose membrane (Thermo) and blocked (Odyssey blocking buffer, Li-cor). The primary antibody was either anti-hERG1 (Alomone APC-109; 1:1000), anti-Hsp47 (MyBioSource MBS 9208399; 1:1000), anti-calnexin (Cell Signaling, 2679), or anti-CD9 (Abcam 65230; 1:500). The secondary antibody was tagged with infrared dye (anti-rabbit 680 (A21076), anti-rabbit 750 (A21039), or anti-mouse 750 (A21037) Alexa Fluor conjugated antibody (Life Technologies; 1:10,000). Membranes were scanned using Odyssey CLx (Li-cor). The incubation period for each primary antibody was overnight (16 hours) in a rocker-type shaker at 70 rpm and 4°C. The secondary antibodies were incubated for 1 hour at room temperature under the same agitation conditions [33].”.
- Page 5, Flow cytometry: Please remove the DOI number for reference and add the reference number for citation.
Answer. We appreciate this important suggestion. It was corrected in the new version of the manuscript (Please the Ref 35, in the page 7).
- References: The reference format should be revised and please change the format according to MDPI reference format.
Answer. We appreciate this important comment. The entire manuscript was revised, and the references were formatted according to MDPI.
- Comments on the Quality of English Language. English languages are fine and minor English editing are required.
Answer. We appreciate this important comment that certainly strengthens our study. The entire manuscript was revised and edited by English Editing MDPI Service (Please, see certificate).

Reviewer 2 Report
Comments and Suggestions for Authors
The aim of the study was to see if the levels of certain protein contents of sEVs isolated from patients’ peripheral blood could be used as a biomarker for diagnosing the different forms and extent of cardiac diseases. Both in vitro and in vivo (i.e. human studies) approaches were employed. Although there are some intriguing data, the manuscript has many weaknesses as outlined below.
1. Except for the discussion, the quality of writing is sub-standard. Certain statements are confusing, some terminologies are used incorrectly, methods are inadequately described, and some data are overinterpreted. The author who wrote the discussion should go over the entire manuscript carefully in order to improve the quality of writing.
2. There are logistic problems. The authors chose two proteins that are “highly expressed in the heart” as the target molecules for the analyses of circulating sEV contents. Why? Please explain the logic behind this decision. Why preselect targets? Would it not be better to do a high-throughput analysis and find a novel biomarker? Since the authors show in vitro data (Figure 2) that production of sEVs containing the preselected targets by cultured cardiomyocytes, they seem to suggest (without saying so) that a portion of sEVs in blood come from cardiomyocytes. Has this been established? It is possible that sEV-hERG1 and sEV-Hsp47 come from blood cells and/or endothelial cells which could be affected by hypoxia and other CVD-related pathological conditions. It has been reported that 99.8% of circulating EVs are generated by hematopoietic cells and that only 0.2% are derived from various tissues. How do sEVs made by cardiomyocytes get into blood? Because the entire circulatory system including the heart is lined by endothelial cells, sEVs made by cardiomyocytes must cross the endothelium without being modified. If this biological issue has been resolved, the authors need to discuss and present evidence for this in the introduction. If there is no evidence, then the data in Figure 2 have little meaning with respect to the rest of the study.
3. In Abstract, the authors state, “human cardiomyocytes released sEV with hERG1 (sEV-hERG1) or Hsp47 (sEV-Hsp47) under hypoxia conditions.” This means that there are two types of sEVs: one with ERG1 and another with Hsp47. The study did not show this. It is possible that both proteins are in the same sEV. To distinguish these two possibilities, immuno-EM using gold particles of two different sizes must be performed, for example.
4. Page 2, lines 11-12: “In addition, cardiac stress tests require stress a cardiology laboratory and a cardiologist.” What does this mean?
5. Were AC-16 cells differentiated before use? What is SFB (page 4)? FBS?
6. Figure 1. What kind of sEV samples did you use? Samples from healthy individuals, patients with certain pathology, etc. Were they from a single individual or a pooled sample? Provide information on samples used in various experiments. In fact, it is necessary to provide such information for all the samples analyzed in various figures. B. These data are n of 1. Each frame should contain several sEVs. Then, the staining data must be quantified by counting >100 vesicles for each category. C and D. What is basal signal? Describe.
7. Figure 2. A. How long were cells alive under hypoxia? Does the reduced sEV production due to cell death? B and C. To standardize the data, was the same number of sEVs used for each category? For all the bar graphs, add dots to represent individual data points as you had done in Figure 5.
8. Stress test and cardiac ischemia are used interchangeably. This is incorrect. Since the extent of cardiac ischemia was not measured, this terminology
9. should not be used to describe results. Use “stress test”, not cardiac ischemia, throughout the manuscript.
10. Page 9, the last sentence. The authors state, “healthy individuals (with unknown cardiac pathologies). This means that these healthy individuals did have some cardiac issues, but the exact pathology was unknown. If so, why are they called healthy individuals?
11. Figure 3A. Add dots to the graph to show individual data points. B and C. What kind of samples are they? From the same individual before and after a stress test? Or are they pooled samples? Describe the nature of these samples in full. Are the sEV preparations for B and C the same or different? If different, the size comparison may not mean much.
12. Figure 5A and B. Describe the nature of samples; from one individual or a pooled sample? If they represent one patient for A and one individual for B, it is difficult to make any conclusions based on these data.
13. Page 14. Figure 5D-F should be Figure 6D-F.
14. Figure 6. These comparisons are interesting, but the comparison was made only between the two types of heart failures. Data from healthy (i.e. no heart failure) individuals should be included.
15. Where is Figure 7?
Comments on the Quality of English LanguageWriting is sub-standard, except for the discussion section which reads well.
Author Response
Reviewer 2.
General comments:
The aim of the study was to see if the levels of certain protein contents of sEVs isolated from patients’ peripheral blood could be used as a biomarker for diagnosing the different forms and extent of cardiac diseases. Both in vitro and in vivo (i.e. human studies) approaches were employed. Although there are some intriguing data, the manuscript has many weaknesses as outlined below.
Answer. We appreciate all the relevant reviewer`s suggestions that will undoubtedly improve the quality of our article. It is important to tell that from the editor office gave us only 10 days to respond to the reviewer, so we did not have to run new experiments.
We have elaborated a detailed response to each comment and criticism of the reviewer. Thus, we have incorporated the changes in the new version of the manuscript.
- Except for the discussion, the quality of writing is sub-standard. Certain statements are confusing, some terminologies are used incorrectly, methods are inadequately described, and some data are overinterpreted. The author who wrote the discussion should go over the entire manuscript carefully in order to improve the quality of writing.
Answer: We appreciate the reviewer`s suggestions to improve the quality of the manuscript. The entire manuscript was modify by English Editing MDPI Service. Please see the new version of the manuscript and the English language editing Certificate by MDPI.
- There are logistic problems. The authors chose two proteins that are “highly expressed in the heart” as the target molecules for the analyses of circulating sEVs contents. Why? Please explain the logic behind this decision. Why preselect targets? Would it not be better to do a high-throughput analysis and find a novel biomarker? Since the authors show in vitro data (Figure 2) that production of sEVs containing the preselected targets by cultured cardiomyocytes, they seem to suggest (without saying so) that a portion of sEVs in blood come from cardiomyocytes. Has this been established? It is possible that sEV-hERG1 and sEV-Hsp47 come from blood cells and/or endothelial cells which could be affected by hypoxia and other CVD-related pathological conditions. It has been reported that 99.8% of circulating EVs are generated by hematopoietic cells and that only 0.2% are derived from various tissues. How do sEVs made by cardiomyocytes get into blood? Because the entire circulatory system including the heart is lined by endothelial cells, sEVs made by cardiomyocytes must cross the endothelium without being modified. If this biological issue has been resolved, the authors need to discuss and present evidence for this in the introduction. If there is no evidence, then the data in Figure 2 have little meaning with respect to the rest of the study.
Answer: We appreciate the reviewer`s suggestions to clarify the criteria for studying the relation between selected proteins in the small extracellular vesicles (sEVs) and cardiovascular diseases.
To make a long history short, in the beginning we found that the transcription factor NFAT5 was increased by hypoxia (in vitro) and ischemia-reperfusion in the kidney (in vivo). We decided to explore if this protein can be a good candidate as a biomarker of cardiac ischemia. As expected, we did not find it free in blood samples. Thus, we decided to explore NFAT5 abundance in the enriched sEVs fraction by ultracentrifugation using human blood samples. Unfortunately, we fail to detect NFAT5 in the sEVs from human blood samples.
In this stage of research, we decide in the next step to study by Western blot using an enriched fraction of sEV from human samples to establish the abundance of different proteins considering (i) highly expressed in the heart and (ii) protein expressed in the membrane of the extracellular vesicles, avoiding sEVs lysis steps.
After intensive research, we have found several candidates and we found that hERG1 and Hsp47 were the most important ones. Using a pool of plasma (EDTA) blood samples obtained from positive cardiac ischemia (confirmed by coronary angiogram). We found by Western blot the presence of hERG1 and Hsp47. In addition, using whole sEVs we found by TEM and Flow cytometer positive signal from hERG1 and Hsp47. Then, using the human cardiomyocyte cell line, we were able to verify the expression of the hERG-1 y Hsp47 proteins in the extracellular vesicles fraction using an ELISA assay, capturing the sEV with Hsp47 and reading CD63 (sEV marker) or capturing the sEV with hERG1 and reading with CD81 (sEV marker). From here, the data suggested that hERG1 and Hsp47 were the most important candidates.
We were very surprised to find Hsp47 into membrane of sEVs. Our hypothesis to propose the presence of Hsp47 in the membrane of sEVs was due to previously it was documented that the stress-inducible Heat shock protein 70 (Hsp70) is expressed in the cell surface of the tumor cell and released into the blood in small extracellular vesicles. The sequence TKD (TKDNNLLGRFELSG) was identified as associated with the extravesicular domain. Thus, we explored the TKD sequence in the Hsp47 protein and found that the sequence “189TKDVERTDGAL200” was present in the protein. Using a commercial antibody against human Hsp47 that recognizes the mentioned sequence, the expression of Hsp47 in sEV was studied. Purified sEVs by ultracentrifugation showed that Hsp47 was associated with CD9 by Western blot analysis (Figure 1A). In addition, the TEM (Figure 1B) and flow cytometer (Figure 1D) experiments produced positive signals for the whole sEVs, suggesting that the sequence “TKDVERTDGALL” of Hsp47 is associated with the extravesicular domain of sEVs.
In the first paragraph in the result section of the submitted manuscript, we mentioned partially the above information according to the next sentence (Please, see the page 8): “We studied the abundance of sEVs proteins by Western blot using an enriched fraction of nanoparticles from human samples, considering they are (i) highly expressed in the heart and (ii) protein is expressed in the membrane of the extracellular vesicles, avoiding steps related to the lysis of sEVs. Thus, the platelet-free plasma (PFP) was prepared from human blood samples within the first hour of the blood sampling. Then, the extracellular vesicles were enriched by ultracentrifugation in a pooled sample from the participants with a positive diagnosis of cardiac ischemia (verified by coronary angiogram), as described in Table 1”.
We agree with the reviewer that the sEV in blood should be from different kinds of cells, making the hematopoietic cells the most important. Here we provide information that the human cardiomyocyte cell line releases these two proteins in the membrane of sEV in-vitro normoxia condition (21% O2) and is induced by 1 hour of hypoxia (1% O2), suggesting to the heart is one of the cell types involved in the secretion of sEV with hERG1 and Hsp47.
We have described in the discussion section this information in the new version of the manuscript with the next sentence (Please see pag. 18 and paragraph 5): “The sources of sEVs in the blood should be from different kinds of cells, with the main contribution from hematopoietic cells, with only a few being derived from a different tissue. Here, we provide evidence that the human cardiomyocyte cell line releases these two proteins in the membrane of sEVs under “in vitro” normoxia conditions (21% O2) and induced after 1 hour of exposure to hypoxia (1% O2), signaling to the heart the cell type involved in the secretion of sEVs containing hERG1 and Hsp47. So far, we do not understand physiological or physiopathological reasons for the secretion of sEV-hERG1 and sEV-Hsp47. We are conducting experiments to understand the role of sEV-hERG1 and sEV-Hsp47 in the mechanisms of cardiovascular diseases and their potential as a biomarkers for this kind of disease”.
- In Abstract, the authors state, “human cardiomyocytes released sEV with hERG1 (sEV-hERG1) or Hsp47 (sEV-Hsp47) under hypoxia conditions.” This means that there are two types of sEVs: one with ERG1 and another with Hsp47. The study did not show this. It is possible that both proteins are in the same sEV. To distinguish these two possibilities, immuno-EM using gold particles of two different sizes must be performed, for example.
Answer. The reviewer is right. We did not show evidence that there are two types of sEV: one with ERG1 and another with Hsp47. Thus we changed the information in the abstract considering the next sentence (Please, see the page 1): “In addition, upon exposure to hypoxia, rat primary cardiomyocytes released sEVs into the extracellular media, and human cardiomyocytes released sEVs containing hERG1 (sEV-hERG1) and/or Hsp47 (sEV-Hsp47)”.
The proposed experiment to use immuno-EM using gold particles of two different sizes to know the population of sEV with hERG1 and Hsp47 is correct, and interesting from a scientific point of view. However, that experiment is a big and long one and the new information did not change the main aspect of the manuscript, suggesting that one of these two candidates is a potential biomarker of cardiovascular research.
We ask the reviewer to consider not including this information in this manuscript.
- Page 2, lines 11-12: “In addition, cardiac stress tests require stress a cardiology laboratory and a cardiologist.” What does this mean?
Answer. In the Page 2 of the new version of the manuscript the sentence was deleted. A new paragraph was included in the new version of the manuscript (Please, see page 2 and paragraph 2: “In addition, cardiac stress tests require a cardiologist trained in the use of cardiology equipment in a laboratory setting”.
- Were AC-16 cells differentiated before use? What is SFB (page 5)? FBS?
Answer. Thank you for your interesting question. The AC-16 cells were not differentiated before use it. In page 4 the SFB was changed by FBS.
- Figure 1. What kind of sEV samples did you use? Samples from healthy individuals, patients with certain pathology, etc. Were they from a single individual or a pooled sample? Provide information on samples used in various experiments. In fact, it is necessary to provide such information for all the samples analyzed in various figures. These data are n of 1. Each frame should contain several sEVs. Then, the staining data must be quantified by counting >100 vesicles for each category. C and D. What is basal signal? Describe.
Answer. We appreciate the reviewer`s suggestions. We increase the information in each part of the figure1:
Figure 1A. The samples used for the analysis by western blot were pooled samples from positive diagnostics of cardiac ischemia described in Table 1. A new sentence was added in the results section of the manuscript (Please, see page 8): “Then, the extracellular vesicles were enriched by ultracentrifugation in a pooled sample from the participants with a positive diagnosis of cardiac ischemia (verified by coronary angiogram), as described in Table 1”.
Figure 1B. The samples used for the analysis by TEM were pooled samples from positive diagnostics of cardiac ischemia described in Table 1. The TEM experiment was conducted to demonstrate the presence of the hERG1 and Hsp47 in the membrane of the extracellular vesicles and it did not have the goal to quantify the positive sEV expressing the hERG1 or Hsp47 proteins. Because we do not have the TEM facilities in our institution, we have limited time and money to use it longer and we decided to take pictures only for isolated sEV.
We ask the reviewer to consider not including this additional information in the manuscript.
A new sentence was added in the results section of the manuscript (Please see page 9 and paragraph 1): “Then, we verified the expression of hERG1 in whole sEVs obtained from a pool of samples from participants with a positive diagnosis of cardiac ischemia, by transmission electron microscopy (TEM) and flow cytometry using the same anti-hERG1 antibody”.
Figure 1C. A new sentence was added in the results section of the manuscript (Please, see page 9): “In addition, using synthetic beads to capture the whole sEV we observed a positive signal in the system when the anti-hERG1 antibody was included in the reaction (Figure 1C, see the red signal)”.
- Figure 2. How long were cells alive under hypoxia? Does the reduced sEV production due to cell death? B and C. To standardize the data, was the same number of sEVs used for each category? For all the bar graphs, add dots to represent individual data points as you had done in Figure 5.
Answer. We appreciate the reviewer`s suggestions to improve the information in Figure 2. We include more information according to:
Figure 2A. We did not measure the mortality of cardiomyocyte primary cell culture associated with hypoxia. Thus, to compare the concentration of sEV between different assays we normalized the total sEV by the total protein amount in the respective plate. A new sentence in the result section was included in the new version of the manuscript (please, see page 10, paragraph 2): "The concentration of sEV was normalized with the total protein amount present in each plate ".
Figure 2B and 2C. We use the volume from culture media equivalent to 1x109 sEV measured by NTA. A new sentence in the method section in the ELISA description was included in the new version of the manuscript: "Then, 100 µL DPBS or a volume of culture media equivalent to 1x109 sEVs and diluted in DPBS from cardiomyocytes exposed to normoxia and hypoxia was incubated for one hour at room temperature and 120 rpm in a rocking system".
The dots to represent individual data points were included in the bar graphs.
- Stress test and cardiac ischemia are used interchangeably. This is incorrect. Since the extent of cardiac ischemia was not measured, this terminology should not be used to describe results. Use “stress test”, not cardiac ischemia, throughout the manuscript.
Answer. The reviewer is right. We changed the term Stress test instead of cardiac ischemia in the new version of the manuscript. The diagnosis of positive or negative diagnosis of cardiac ischemia was done by a team of cardiologists headed by Dr. Ricardo Larrea from the Cardiology Department of the Dávila Clinic and the positive diagnosis was confirmed by a Coronary angiogram. Thus, we do not know the level of cardiac ischemia but we are sure of the positive condition of heart ischemia in the enrolled participants.
A new sentence in the method section was included in the new version of the manuscript (Please, see page 4): “The determination of a positive or negative diagnosis of cardiac ischemia was conducted by a team of cardiologist, headed by Dr. Ricardo Larrea, from the Cardiology Department of the Clínica Dávila, and a positive diagnosis was confirmed by coronary angiogram”.
- Page 9, the last sentence. The authors state, “healthy individuals (with unknown cardiac pathologies). This means that these healthy individuals did have some cardiac issues, but the exact pathology was unknown. If so, why are they called healthy individuals?
Answer. The reviewer is right and we eliminated the confusing phrase. A new sentence in the results section was included in the new version of the manuscript (Please, see page 12): “We sampled healthy participants in resting conditions (i.e., without exercise)”.
- Figure 3A. Add dots to the graph to show individual data points. B and C. What kind of samples are they? From the same individual before and after a stress test? Or are they pooled samples? Describe the nature of these samples in full. Are the sEV preparations for B and C the same or different? If different, the size comparison may not mean much.
Answer. We appreciate the reviewer`s suggestions to improve the information in Figure 3. We include more information according to:
Figure 3A. The dots to represent individual data points were included in the bar graphs.
Figure 3B and 3C. A new sentence in the Figure 3 legend was included in the new version of the manuscript: " In (B) and (C), the amounts of 1x109 sEVs were determined for each blood plasma (PFP) sample and immunolabeled for 1 hour with primary antibody and then 1 hour with Alexa Fluor 532 secondary antibody ".
- Figure 5A and B. Describe the nature of samples; from one individual or a pooled sample? If they represent one patient for A and one individual for B, it is difficult to make any conclusions based on these data.
Answer. We appreciate the reviewer’s comment. A new sentence in the results section was included in the new version of the manuscript (Please, see page 14): “In addition, the characterizations of sEV-hERG1 and sEV-Hsp47 in the samples from participants with CHF (n=10) and DHF (n=10) were conducted using a flow cytometer (a representative image is shown in Figure 5 A-B). Analyzing the results of the flow cytometry, we found that both groups of participants with CHF and DHF showed detectable levels of sEV-hERG1 and sEV-Hsp47”.
- Page 14. Figure 5D-F should be Figure 6D-F
Answer. We apologize for this unfortunate error in the article writing. The information was corrected in the new version of the manuscript.
- Figure 6. These comparisons are interesting, but the comparison was made only between the two types of heart failures. Data from healthy (i.e. no heart failure) individuals should be included.
Answer. We appreciate the reviewer’s comment. Considering that this is an exploratory study in the heart failure population (n=10 compensated heart failure and n=10 decompensated heart failure), we did not run experiments to compare healthy and heart failure individuals, because the heart failure individuals are more comparable in terms of clinical condition and therapeutics aspect. We included a new paragraph in the discussion section in the new version of the manuscript (Please, see page 18): “The results are very interesting regarding the two types of heart disease. In the stress test participants, we found that when transient cardiac ischemia is provoked, blood levels of sEV-hERG1 and sEV-Hsp47 significantly increased after 15 min of finishing the test. On the other hand, in the heart failure participants, the levels of total sEV and the sEVs containing hERG1 and Hsp47 decreased in the samples from participants with decompensated heart failure compared with compensated heat failure. Moreover, the relationship between the levels of sEV-hERG1 and total sEVs decreased in the decompensated heart failure group compared with the compensated group. Similar results were observed for the levels of sEV-Hsp47. The stress test induces an acute cardiac condition, while heart failure is a chronic condition. Thus, we considered it appropriate to only compare within each group because of the evident clinical differences between both groups. In this way, as mentioned above, we are preparing an ELISA kit to carry out a clinical study with a robust number of samples to be able to compare different cardiovascular conditions. Here, we only analyzed the individuals in each group. Thus, in the present state of the research, it is not recommended to compare both kinds of cardiovascular conditions”.
- Where is Figure 7?
Answer. We apologize for this unfortunate error in the article edition. The information was corrected in the new version of the manuscript (Please, see page 18).
- Comments on the Quality of English Language
Writing is sub-standard, except for the discussion section which reads well.
Answer. The article was edited by the company suggested by the IJMS (English Editing MDPI Service) and we hope the new version of the manuscript has the standard to be published (Please, see the certificate).
Reviewer 3 Report
Comments and Suggestions for Authors
The authors’ hypothesis points on the identification of circulating EVs displaying discrete surface profiles in CVD. The study is of interest, however data reported are confusing particularly in the description of population differently evaluated for the expression of hERG1 an Hsp47
Comments
The methodology used to search for overexpressed proteins must be included.
Are from healthy subjects EVs characterized and reported in Fig.1
The expression of specific proteins by TEM is low. More than 1 EV must be shown. Moreover, the number of EVs used for WB must be included.
A negative EV marker is required by the ISEV guidelines.
Fig.S1 a EV marker must be reported
Since cardiomyocytes are expected to release EVs enriched in hERG1, Hsp47 the cell of origine of these enriched EVs must be shown.
Again Fig. 3 must also show the most relevant cells of origin.
Extracellular vesicles with hERG-1 and Hsp47 decreased in the blood of participants with
decompensated heart failure. In this paragraph the Fig. numbers do not correspond.
The authors must explain why the DHF have a reduced number of EVs. This observation does not fit with the results of ischemic patients.
The authors suggest that the enrichment of EV-ERG1 may represent a protective mechanisms. Again why EV-ERG1 are decreased in DHF?
The study is of interest, however data reported are confusing particularly in the description of population differently evaluated for the expression of hERG1 an Hsp47.
Author Response
Reviewer 3.
Comments and Suggestions for Authors
The authors’ hypothesis points on the identification of circulating EVs displaying discrete surface profiles in CVD. The study is of interest, however data reported are confusing particularly in the description of population differently evaluated for the expression of hERG1 an Hsp47.
Answer. We appreciate all the relevant reviewer`s suggestions that will undoubtedly improve the quality of our article. It is important to tell that from the editor office gave us only 10 days to respond to the reviewer, so we did not have to run new experiments.
We have elaborated a detailed response to each comment and criticism of the reviewer. Thus, we have incorporated the changes in the new version of the manuscript.
- The methodology used to search for overexpressed proteins must be included.
Answer. We appreciate the reviewer`s suggestions to clarify the criteria for studying the relation between selected proteins in the small extracellular vesicles (sEV) and cardiovascular diseases.
To make a long history short, in the beginning, we found that the transcription factor NFAT5 was increased by hypoxia (in vitro) and ischemia-reperfusion in the kidney (in vivo) [PLoS One. 2012;7(7):e39665. doi: 10.1371/journal.pone.0039665]. We decided to explore if this protein can be a good candidate as a biomarker of cardiac ischemia. As expected, we did not find it free in blood samples. Thus, we decided to explore NFAT5 abundance in the enriched sEV fraction by ultracentrifugation using human blood samples. Unfortunately, we fail to detect NFAT5 in the sEV from human blood samples.
In this stage of research, we decide in the next step to study by Western blot using an enriched fraction of sEV from human samples to establish the abundance of different proteins considering (i) highly expressed in the heart and (ii) protein expressed in the membrane of the extracellular vesicles, avoiding sEV lysis steps.
After intensive research, we have found several candidates and we found that hERG1 and Hsp47 were the most important ones. Using a pool of plasma (EDTA) blood samples obtained from positive cardiac ischemia (confirmed by coronary angiogram). We found by Western blot the presence of hERG1 and Hsp47. In addition, using whole sEVs we found by TEM and Flow cytometer positive signal from hERG1 and Hsp47. Then, using the human cardiomyocyte cell line, we were able to verify the expression of the hERG-1 y Hsp47 proteins in the extracellular vesicles fraction using an ELISA assay, capturing the sEVs with Hsp47 and reading CD63 (sEV marker) or capturing the sEVs with hERG1 and reading with CD81 (sEVs marker). From here, the data suggested that hERG1 and Hsp47 were the most important candidates.
We were very surprised to find Hsp47 into the membrane of sEVs. Our hypothesis to propose the presence of Hsp47 in the membrane of sEVs was due to previously it was documented that the stress-inducible Heat shock protein 70 (Hsp70) is expressed in the cell surface of the tumor cell and released into the blood in small extracellular vesicles. The sequence TKD (TKDNNLLGRFELSG) was identified as associated with the extravesicular domain. Thus, we explored the TKD sequence in the Hsp47 protein and found that the sequence “189TKDVERTDGAL200” was present in the protein. Using a commercial antibody against human Hsp47 that recognizes the mentioned sequence, the expression of Hsp47 in sEVs was studied. Purified sEV by ultracentrifugation showed that Hsp47 was associated with CD9 by Western blot analysis (Figure 1A). In addition, the TEM (Figure 1B) and flow cytometer (Figure 1D) experiments produced positive signals for the whole sEVs, suggesting that the sequence “TKDVERTDGALL” of Hsp47 is associated with the extravesicular domain of sEVs.
In the first paragraph in the result section of the new version of the manuscript we mentioned partially the above information according to the next sentence “We studied the abundance of sEV proteins by Western blot using an enriched fraction of nanoparticles from human samples, considering they are (i) highly expressed in the heart and (ii) protein is expressed in the membrane of the extracellular vesicles, avoiding steps related to the lysis of sEVs. Thus, the platelet-free plasma (PFP) was prepared from human blood samples within the first hour of the blood sampling. Then, the extracellular vesicles were enriched by ultracentrifugation in a pooled sample from the participants with a positive diagnosis of cardiac ischemia (verified by coronary angiogram), as described in Table 1. Using the anti-hERG1 antibody, which recognizes the extracellular domain (430AFLLKETEEGPAPATE445) of hERG1, in the Western blot analysis we found that hERG1 was present in the sEVs and associated with CD9 (a marker of sEVs) (Figure 1A)”.
”.
We have described in the discussion section this information in the new version of the manuscript with the next sentence: “ The sources of sEVs in the blood should be from different kinds of cells, with the main contribution from hematopoietic cells, with only a few being derived from a different tissue. Here, we provide evidence that the human cardiomyocyte cell line releases these two proteins in the membrane of sEVs under “in vitro” normoxia conditions (21% O2) and induced after 1 hour of exposure to hypoxia (1% O2), signaling to the heart the cell type involved in the secretion of sEVs containing hERG1 and Hsp47. So far, we do not understand physiological or physiopathological reasons for the secretion of sEV-hERG1 and sEV-Hsp47. We are conducting experiments to understand the role of sEV-hERG1 and sEV-Hsp47 in the mechanisms of cardiovascular diseases and their potential as a biomarkers for this kind of disease. ” (Please, see page 21).
- Are from healthy subjects EVs characterized and reported in Fig.1.
Answer. We appreciate the reviewer`s suggestions. We increase the information in each part of the figure1:
Figure 1A. The samples used for the analysis by western blot were pooled samples from positive diagnostics of cardiac ischemia described in Table 1. A new sentence was added in the results section of the manuscript (Please, page 84 and paragraph 4): “ Then, the extracellular vesicles were enriched by ultracentrifugation in a pooled sample from the participants with a positive diagnosis of cardiac ischemia (verified by coronary angiogram), as described in Table 1”.
Figure 1B. The samples used for the analysis by TEM were pooled samples from positive diagnostics of cardiac ischemia described in Table 1. The TEM experiment was conducted to demonstrate the presence of the hERG1 and Hsp47 in the membrane of the extracellular vesicles and it did not have the goal to quantify the positive sEV expressing the hERG1 or Hsp47 proteins. Because we do not have the TEM facilities in our institution, we have limited time to use them and we concentrate in to take pictures only for isolated sEV.
We ask the reviewer to consider not including this additional information in the manuscript.
A new sentence was added in the results section of the manuscript (Please see page 7): “ Then, we verified the expression of hERG1 in whole sEVs obtained from a pool of samples from participants with a positive diagnosis of cardiac ischemia, by transmission electron microscopy (TEM) and flow cytometry using the same anti-hERG1 antibody.”.
Figure 1C. A new sentence was added in the results section of the manuscript: “In addition, using synthetic beads to capture the whole sEV we observed a positive signal in the system when the anti-hERG1 antibody was included in the reaction (Figure 1C, see the red signal)”.
- The expression of specific proteins by TEM is low. More than 1 EV must be shown.
Answer. We appreciate the reviewer`s suggestions. Figure 1B. The samples used for the analysis by TEM were pooled samples from positive diagnostics of cardiac ischemia described in Table 1. The TEM experiment was conducted to demonstrate the presence of the hERG1 and Hsp47 in the membrane of the extracellular vesicles and it did not have the goal to quantify the positive sEV expressing the hERG1 or Hsp47 proteins. Because we do not have the TEM facilities in our institution, we have limited time to use them, and we concentrate in to take pictures only for isolated sEV.
We ask the reviewer to consider not including this additional information in the manuscript.
A new sentence was added in the results section of the manuscript (Please see page 9, paragraph 3): “ Then, we verified the expression of hERG1 in whole sEVs obtained from a pool of samples from participants with a positive diagnosis of cardiac ischemia, by transmission electron microscopy (TEM) and flow cytometry using the same anti-hERG1 antibody. The TEM results showed a positive signal for the hERG1 associated with the membrane of the sEVs (Figure 1B).”.
- A negative EV marker is required by the ISEV guidelines.
Answer. We appreciate the reviewer`s suggestions to increase the data to verify the presence of purified small extracellular vesicles (sEV). A new supplementary figure was included in the new version of the manuscript (Figure S2).
- S1 a EV marker must be reported.
Answer. We appreciate the reviewer`s suggestions. An EV marker was reported in the figure S1.
- Since cardiomyocytes are expected to release EVs enriched in hERG1, Hsp47 the cell of origine of these enriched EVs must be shown.
Answer. We did not take pictures of the primary cell culture of cardiomyocytes used in the present research. The protocol to purify the cardiomyocyte was described by the authors Danna Morales and Diego Varela and the primary cell culture protocol was well described in the publication “Calcium-dependent inactivation controls cardiac L-type Ca2+ currents under β-adrenergic stimulation. J Gen Physiol 2019. 151:786–797. doi: 10.1085/jgp.201812236”. It is reference 29 in the new version of the manuscript.
Thus, we ask you to review do not consider the cardiomyocyte picture in the new version of the manuscript.
- Again Fig. 3 must also show the most relevant cells of origin.
Answer. A picture of AC-16 cells was included in the supplementary Figure S3.
- Extracellular vesicles with hERG-1 and Hsp47 decreased in the blood of participants with decompensated heart failure. In this paragraph the Fig. numbers do not correspond.
Answer. We apologize for the mistake and appreciate this important comment. A phrase was missing during the last edition. In the new version of the manuscript, it was re-written as “ “Given these results, we analyzed the correlations between sEV/sEV-hERG1 and sEV/sEV-Hsp47 and the total concentrations of EV and EV-Hsp47 for each group of patients (Figure 7).”.
- The authors must explain why the DHF have a reduced number of EVs. This observation does not fit with the results of ischemic patients.
Answer. We appreciate the interesting comments of the reviewer about the participants with different cardiovascular diseases. We included a new paragraph in the discussion section in the new version of the manuscript (Please, see page 21, last paragraphs: “ “The results are very interesting regarding the two types of heart disease. In the stress test participants, we found that when transient cardiac ischemia is provoked, blood levels of sEV-hERG1 and sEV-Hsp47 significantly increased after 15 min of finishing the test. On the other hand, in the heart failure participants, the levels of total sEV and the sEVs containing hERG1 and Hsp47 decreased in the samples from participants with decompensated heart failure compared with compensated heart failure. Moreover, the relationship between the levels of sEV-hERG1 and total sEV decreased in the decompensated heart failure group compared with the compensated group. Similar results were observed for the levels of sEV-Hsp47. The stress test induces an acute cardiac condition, while heart failure is a chronic condition. Thus, we considered it appropriate to only compare within each group because of the evident clinical differences between both groups. In this way, as mentioned above, we are preparing an ELISA kit to carry out a clinical study with a robust number of samples to be able to compare different cardiovascular conditions. Here, we only analyzed the individuals in each group. Thus, in the present state of the research, it is not recommended to compare both kinds of cardiovascular conditions.”.
- The authors suggest that the enrichment of EV-ERG1 may represent a protective mechanisms. Again, why EV-ERG1 are decreased in DHF?
Answer. We do not have evidence to suggest that sEV-ERG1 has protective mechanisms. Thus, in the new version of the manuscript, we deleted that paragraph from the discussion section. In addition, we do not have data to explain why sEV-ERG1 is decreased in DHF compared with the CHF. More research is necessary to explain these findings.
- The study is of interest, however data reported are confusing particularly in the description of population differently evaluated for the expression of hERG1 an Hsp47.
Answer. We appreciate the interesting comments of the reviewer about the different participants with cardiovascular diseases. Once again, we included a new paragraph in the discussion section in the new version of the manuscript (Please, see page 21): “The results are very interesting regarding the two types of heart disease. In the stress test participants, we found that when transient cardiac ischemia is provoked, blood levels of sEV-hERG1 and sEV-Hsp47 significantly increased after 15 min of finishing the test. On the other hand, in the heart failure participants, the levels of total sEVs and the sEVs containing hERG1 and Hsp47 decreased in the samples from participants with decompensated heart failure compared with compensated heart failure. Moreover, the relationship between the levels of sEV-hERG1 and total sEVs decreased in the decompensated heart failure group compared with the compensated group. Similar results were observed for the levels of sEV-Hsp47. The stress test induces an acute cardiac condition, while heart failure is a chronic condition. Thus, we considered it appropriate to only compare within each group because of the evident clinical differences between both groups. In this way, as mentioned above, we are preparing an ELISA kit to carry out a clinical study with a robust number of samples to be able to compare different cardiovascular conditions. Here, we only analyzed the individuals in each group. Thus, in the present state of the research, it is not recommended to compare both kinds of cardiovascular conditions”.
Round 2
Reviewer 1 Report
Comments and Suggestions for Authors
General Comments
The manuscript “Levels of Small Extracellular Vesicles Containing hERG-1 and Hsp47 as Potential Biomarkers for Cardiovascular Diseases” is significantly improved after the authors' corrections. The authors have appropriately responded to the original concerns. I think the changes are acceptable. I recommend the publication of the manuscript.
Author Response
Answer. We appreciate the recommendation from the reviewer to publish the present version of the manuscript.
Reviewer 2 Report
Comments and Suggestions for Authors
The authors responded to my comments only partially; they delt with those that are easy to address, leaving the major criticisms unrectified. Writing is much better although the style is wordy and repetitious. In addition, too many sentence connectors are used, especially “thus”. The authors may wish to delete most of them, leaving only those that are absolutely necessary. Because the writing is clearer, it has revealed some issues in some experimental approaches and data interpretations, and some controls are sorely needed. Overall, the patient studies are interesting and the paper seems to show correlation between the blood sEV, sEV-hERG1 and sEV-Hsp47 levels and certain types of CVD. One critical issue is the origin of these sEVs. The fact that the authors used cultured cardiomyocytes to show their ability to make sEVs, especially under the hypoxic condition seems indicate their unstated suggestion that the increased presence of blood sEV, sEV-hERG1 and sEV-Hsp47 is due to increased production of these sEVs by cardiomyocytes. This is indeed one of the possibilities, but the way the paper is written is vague and gives a sense of trickery. The authors should discuss this possibility in a more straightforward and logical manner, citing appropriate papers and providing biologically feasible arguments. This will require some work, but it will be worth the effort. I strongly suggest this so that the in vitro and clinical data may be logically tied. If this is not done, I suggest removing in vitro data, or at least move them to supplement and reduce the level of emphasis in the text. Specific issues are outlined below.
1. Abstract. Line 21. Change “in the membrane of extracellular vesicles (sEVs)” to “in the membrane of small extracellular vesicles (sEVs)”. Line 34. Remove “Furthermore”.
2. Line 84. Change “membrane-contained vesicles” to “membrane-bound vesicles”.
3. Lines 87-88. “The presence of sEVs has been described in different vascular and cardiac diseases.” State where these sEVs are present. In blood, vascular wall, cardiac tissue, etc.
4. Line 107. “the KCNH2 gene encodes an ion channel involved in the rapid component of IKr.” This sentence does not make sense. Rewrite.
5. Line 114. Remove “Thus”.
6. Line 116. Change “to identify” to “as”.
7. Line 125. The authors state that for the stress test, “A total of 26 participants were enrolled.” When cardiac ischemia was diagnosed, 50% was positive for cardiac ischemia and the other 50% was negative. This could happen by chance, but I find it extremely unlikely. I am wondering if cardiac stress was given to more than 26 individuals, and out of such a group, 13 positive and 13 negative cases were picked up. If this were the case, the description given in the Method is misleading. Please describe how the exact 1:1 case ratio was achieved, and if the two groups were selected from a larger pool, were they selected randomly? If not randomly selected, what were the criteria for selection. If there were more than 26 individual who participated in the cardiac stress study, all the data should be included in the study. If the 26 subjects were subjectively selected, the data should be thrown out, or justify logically why that was necessary. This issue is critical as a large portion of the data comes from these participants.
8. Line 133. For the heart failure study, a total of 20 patients were chosen. Once again, the ratio between CHF and DHF is 1:1. It is likely that the authors selected 10 patients for each case from a larger population of patients. If so, how were those 20 patients selected? Randomly from two piles of patients? As I pointed out in the comment above, how exactly these 20 cases were chosen must be clearly described in a scientifically acceptable manner (i.e. randomness of selection is established).
9. Line 220. Since this is the first place where antibodies are described, and both anti-hERG1 and anti-Hsp47 are made in rabbits, “antibody” at the beginning of the line should be changed to “anti-rabbit IgG”.
10. Lines 225 and 231. Remove “Extracellular” from the title, or replace it with “sEV” or “Small EV”.
11. Lines 241-242. Rewrite “the migration front crossed the gel concentrator–gel separator junction” as “the dye front crossed the junction between the concentrator and separator gels.”
12. Line 248. “or anti-mouse 750 (A21037) Alexa Fluor conjugated”. As far as I can tell, all the primary antibodies are made in rabbits. Why use anti-mouse? This will not work.
13. Line 255. For immunogold labeling, sEVs were fixed “with 2% osmium tetroxide” and also negatively stained later. However, the EM micrographs do not show membrane. Why? Please explain. Showing the membrane is critical for establishing the gold label is outside the vesicle.
14. Lines 270-272. “The pellets were resuspended in 10% w/v bovine serum albumin (BSA; Winkler Ltda., Santiago, Chile, Cat. 271 BM-0150) for 45 min at RT. Then, the pellet was resuspended in a 2% BSA solution…” How is it possible to resuspended the pellet that is already resuspend? Was there a centrifugation step after 45 min incubation?
15. Line 274. “or the isotype control mouse IgG1 (BD Biosciences, Cat. 349040)”. Why mouse IgG? Since all the primary antibodies are rabbit antibodies, this control is meaningless for non-specific binding of rabbit Ig. Redo the experiments.
16. Lines 277-278. “secondary antibody a-mouse IgG1 Alexa Fluor 488 (BioLegend, Cat. 406626) for 30 min at RT.” The primary anti-hERG1 and anti-Hsp47 are rabbit antibodies. These experiments should not have worked. Using the proper animal combination (i.e. rabbit primary and anti-rabbit secondary), please redo the experiments.
17. Line 295. “1ng/µ”???
18. Line 303. Change “spectrophotometer with an absorbance of 450 nm” to “spectrophotometer for an absorbance at 450 nm.”
19. Line 305. “The data are expressed as the mean ± SEM.” This may not be true as some data appear to be expressed as mean ± SD. Please check, and correct as necessary.
20. Line 309. Tittle. Change “hERG1 Is Expressed…” to “. hERG1 and Hsp47 are Expressed…” This change is necessary because Hsp47 expression is also described in this section. This change requires rewriting parts of this section. Please go over carefully so as to discuss both proteins.
21. Lines 312-313. “(ii) protein is expressed in the membrane of the extracellular vesicles, avoiding steps related to the lysis of sEVs.” In this sentence, the meaning of “avoiding steps related to the lysis of sEVs” is unclear. Delete this phrase.
22. Line 319. “hERG1 was present in the sEVs and associated with CD9 (a marker of sEVs).” This is a misleading statement. The data do not show that hERG1 is associated with CD9. In general, association of two molecules means binding of the two, hence this statement is an overinterpretation of the data. Change this sentence to “hERG1 was present in the sEVs as identified by CD9.”
23. Line 324. Figure 1B. Here sEV membrane association of hERG1 (and later also Hsp47) is described. As I commented in my first review, this figure represents n=1 data. Such an illustration, no matter how nice looking it is, has little scientific value as such data could be selected or the labeling (or non-labeling) is accidental. This is a direct visualization that shows hERG1 and Hsp47 on sEV, and as such, it is a very important direct observation. However, showing only one example is not convincing. Please show several such cases. Also, it is good to show some statistical analyses (such as % of labeled sEVs out of >100 vesicles counted). The authors stated in their response that such demands cannot be readily met. If the presentation of these EM data cannot be scientifically improved, they should be deleted (as n=1 data has little scientific value), or they may be used as supplemental data. Please also see my comment #13.
24. Line 335. Please describe how 13 positive and 13 negative cases were achieved? Please see my comments #7.
25. Line 354. “Hsp47 was associated with CD9”. This is an overinterpretation of the data. The gel data does not show their binding.
26. Line 368. “cardiomyocytes’”. Remove the apostrophe.
27. Lines 372 and 374. Change “consists in” to “consists of”.
28. Lines 388, 393 and 397. Please confirm if SD is correct as in the methods section, you state that (all) data are presented with SEM.
29. Lines 389, 390, 394, and 398. “Significance is denoted as * p < 0.05 and ** p < 0.005 for normoxia.” Change “for” to either “compared with” or “vs”
30. Lines 393 and 397. “n=3”. There are more than 3 data points. Please correct.
31. Line 399. “Effect of Ischemia in Stress Test Participants on Secretion of sEVs…” This is a misleading title as the human study does not analyze secretion of sEVs. It looks at sEVs in blood, which is not secretion. Again, I see this as a subliminal suggestion that sEVs come from cardiomyocytes which appear to secrete sEVs under the ischemic condition in vitro. Some investigators believe and also have reported that AC16 cells are quite different from in site cardiomyocytes. In vitro data may not reflect what happens in vivo.
32. Line 439. Change “a secondary antibody conjugated with fluorescence” to “a fluorescent secondary antibody”.
33. Lines 442-444. “In this manner, a small volume of PFP (5-10 mL) was used to obtain the total amount of 1x109 sEVs. The PFP samples were diluted to 0.5 mL with DPBS…” Delete “In this manner”. It is not clear how PFP in 5-10 mL can be diluted to make 0.5 mL of sample. Maybe something is not described. Please rewrite. Is 5-10 mL of PFP a small volume?
34. Lines 459-461. “Thus, we speculate that the sEV-hERG1 and sEV-Hsp47 concentrations in the blood samples increased, at least in the heart, in response to the development of ischemia.” What does “at least in the heart” mean? Was the blood sample collected from the heart? Please explain logically here what this phrase means. Unless there is a god explanation, please delete this phrase.
35. Lines 473-474. Are these SBP values shown with SD or SEM? How about the values in Table 2?
36. Figures 5, 6 and 7. In these studies when CHF and DHF are compared, it is not enough to compare just these two. The data from patients must be contrasted also to the data from healthy individuals. This is critical because, for example, it could be possible that the data from DHF patients are closer to those of the healthy individual! So, if diagnosis is one of the aims of this study, normal control must be included in the analysis.
37. Lines 529-531. “we analyzed the correlations between sEVs/sEV-hERG1 and sEVs/sEV-Hsp47 and the total concentrations of EV and EV-Hsp47 for each group of patients.” It is not clear what this sentence means. The figure legend seems to describe the analyses better.
38. Figure 7. The appearance of the two lines in A and B are the same. Please make the two lines distinct (by color, one solid and one dotted, etc.) and indicate what is what.
39. Lines 627-628. “the secretion of hERG1 by extracellular vesicles could be a response mechanism against cardiac damage.” The meaning of this sentence is not clear. Is hERG1 secreted by EVs?
40. Line 636. “Hsp47 myofibroblasts are the primary mediators of tissue fibrosis…” Do you mean “Hsp47 in myofibroblasts is the primary mediator of tissue fibrosis…”?
41. Lines 659-660. “signaling to the heart the cell type involved in the secretion of sEVs containing hERG1 and Hsp47.” The meaning of this phrase is not clear. Please rewrite.
42. Figure 8. The cartoon of the EV shows Hsp47 is embedded in the membrane of the vesicle. As far as I remember, Hsp47 does not have a transmembrane domain. How does it span the membrane as depicted in the figure? Please discuss how this protein can be expressed on the outer surface of sEV. It may be necessary to modify this cartoon.
Comments on the Quality of English LanguageIn many instances, connecting words are used inappropriately. Writing is wordy and repetitious.
Author Response
Round 2. Reviewer 2.
The authors responded to my comments only partially; they delt with those that are easy to address, leaving the major criticisms unrectified. Writing is much better although the style is wordy and repetitious. In addition, too many sentence connectors are used, especially “thus”. The authors may wish to delete most of them, leaving only those that are absolutely necessary. Because the writing is clearer, it has revealed some issues in some experimental approaches and data interpretations, and some controls are sorely needed. Overall, the patient studies are interesting and the paper seems to show correlation between the blood sEV, sEV-hERG1 and sEV-Hsp47 levels and certain types of CVD. One critical issue is the origin of these sEVs. The fact that the authors used cultured cardiomyocytes to show their ability to make sEVs, especially under the hypoxic condition seems indicate their unstated suggestion that the increased presence of blood sEV, sEV-hERG1 and sEV-Hsp47 is due to increased production of these sEVs by cardiomyocytes. This is indeed one of the possibilities, but the way the paper is written is vague and gives a sense of trickery. The authors should discuss this possibility in a more straightforward and logical manner, citing appropriate papers and providing biologically feasible arguments. This will require some work, but it will be worth the effort. I strongly suggest this so that the in vitro and clinical data may be logically tied. If this is not done, I suggest removing in vitro data, or at least move them to supplement and reduce the level of emphasis in the text. Specific issues are outlined below.
Answer. We appreciate the valuable comments from the reviewer. We understand that the suggested changes are necessary to produce better version of the manuscript. In the new document we will make a huge effort to respond to each of the demands.
Regarding the writing style, how was mentioned in past answer, the entire manuscript was modify by English Editing MDPI Services, suggested by the journal. In the new version, we have included some stylistic changes to satisfy the reviewer's demands. Additionally, we incorporated the requested controls.
Regarding to cultured cardiomyocytes data to demonstrate their ability to release sEVs, especially under hypoxic conditions, we are clear that this information only suggests that cardiomyocytes cell line have the biological capability to release EV-hERG1 and EV-Hsp47, but we cannot say that the EV-hERG1 and EV-Hsp47 found it in blood samples from human are coming from the cardiomyocyte belong to the heart. This concept has been modified so that this information is not vague and much less so that there is a sense of trickery. The new version of the manuscript discusses this possibility in a more direct and logical manner, citing appropriate articles and providing biologically viable arguments.
- Abstract. Line 21. Change “in the membrane of extracellular vesicles (sEVs)” to “in the membrane of small extracellular vesicles (sEVs)”. Line 34. Remove “Furthermore”.
Answer. According with reviewer´s suggestion the changes are included in the new version of the manuscript.
- Line 84. Change “membrane-contained vesicles” to “membrane-bound vesicles”.
Answer. According with reviewer´s suggestion the changes are included in the new version of the manuscript.
- Lines 87-88. “The presence of sEVs has been described in different vascular and cardiac diseases.” State where these sEVs are present. In blood, vascular wall, cardiac tissue, etc.
Answer. According with reviewer´s suggestion the changes are included in the new version of the manuscript. The next sentence and a reference was added in line 91: “The presence of circulating sEVs has been described in different cardiovascular diseases."
- Line 107. “the KCNH2 gene encodes an ion channel involved in the rapid component of IKr.” This sentence does not make sense. Rewrite.
Answer. The sentence was Re-wrote to in line 114. “Besides, the voltage-dependent potassium channel a-subunit KV11.1 (also named the human ether-à-go-go-related gene 1 (hERG1)) is essential for normal electrical activity in the heart, regulating the action potential of the heart."
- Line 114. Remove “Thus”.
Answer. According with reviewer´s suggestion the changes are included in the new version of the manuscript.
- Line 116. Change “to identify” to “as”.
Answer. According with reviewer´s suggestion the changes are included in the new version of the manuscript.
- Line 125. The authors state that for the stress test, “A total of 26 participants were enrolled.” When cardiac ischemia was diagnosed, 50% was positive for cardiac ischemia and the other 50% was negative. This could happen by chance, but I find it extremely unlikely. I am wondering if cardiac stress was given to more than 26 individuals, and out of such a group, 13 positive and 13 negative cases were picked up. If this were the case, the description given in the Method is misleading. Please describe how the exact 1:1 case ratio was achieved, and if the two groups were selected from a larger pool, were they selected randomly? If not randomly selected, what were the criteria for selection. If there were more than 26 individual who participated in the cardiac stress study, all the data should be included in the study. If the 26 subjects were subjectively selected, the data should be thrown out, or justify logically why that was necessary. This issue is critical as a large portion of the data comes from these participants.
Answer. According with reviewer´s suggestion the changes are included in the new version of the manuscript, adding the next paragraph in the method section: “We enrolled 120 patients in the study after signed the Informed Consent. In this group, 13 received a positive diagnosis of cardiac ischemia in the Stress Test by the cardiologists from Clnica Dávila. In this way, 13 patients were randomly included with a negative diagnosis of cardiac ischemia in the Stress Test to have the same participants in both groups. The inclusion criterion for Stress Test included: age ≥ 18 years at the time of signing the informed consent.”
Regarding to requesting to include all the data we did not analyze the EV-hERG1 and EV-Hsp47 levels in the entire group. As a mentioned in the discussion section we are preparing an ELISA kit to run a study with a big number of participants to verify this preliminary data.
- Line 133. For the heart failure study, a total of 20 patients were chosen. Once again, the ratio between CHF and DHF is 1:1. It is likely that the authors selected 10 patients for each case from a larger population of patients. If so, how were those 20 patients selected? Randomly from two piles of patients? As I pointed out in the comment above, how exactly these 20 cases were chosen must be clearly described in a scientifically acceptable manner (i.e. randomness of selection is established).
Answer. According with reviewer´s suggestion the changes are included in the new version of the manuscript, adding the next paragraph in the method section. “The cardiologists team at the Dávila clinic invited patients with heart failure who are regularly treated at this institution to participate in this study. Participants were selected at random until there were 10 participants in each group for the purposes of this research.”
- Line 220. Since this is the first place where antibodies are described, and both anti-hERG1 and anti-Hsp47 are made in rabbits, “antibody” at the beginning of the line should be changed to “anti-rabbit IgG”.
Answer. According with reviewer´s suggestion the changes are included in the new version of the manuscript.
- Lines 225 and 231. Remove “Extracellular” from the title, or replace it with “sEV” or “Small EV”.
Answer. According with reviewer´s suggestion the changes are included in the new version of the manuscript.
- Lines 241-242. Rewrite “the migration front crossed the gel concentrator–gel separator junction” as “the dye front crossed the junction between the concentrator and separator gels.”
Answer. According with reviewer´s suggestion the changes are included in the new version of the manuscript.
- Line 248. “or anti-mouse 750 (A21037) Alexa Fluor conjugated”. As far as I can tell, all the primary antibodies are made in rabbits. Why use anti-mouse? This will not work.
Answer. We apologize for this unfortunate error in the article writing. The information was corrected in the new version of the manuscript and the “anti-mouse 750 (A21037)” information was corrected.
- Line 255. For immunogold labeling, sEVs were fixed “with 2% osmium tetroxide” and also negatively stained later. However, the EM micrographs do not show membrane. Why? Please explain. Showing the membrane is critical for establishing the gold label is outside the vesicle.
Answer. The figure 1, with the EM micrographs was deleted of this section and included as a supplementary figure. Why do not have an explication why the EM micrographs do not show membrane. We reviewed the protocol and the information was clarified in the method section of the manuscript.
- Lines 270-272. “The pellets were resuspended in 10% w/v bovine serum albumin (BSA; Winkler Ltda., Santiago, Chile, Cat. 271 BM-0150) for 45 min at RT. Then, the pellet was resuspended in a 2% BSA solution…” How is it possible to resuspended the pellet that is already resuspend? Was there a centrifugation step after 45 min incubation?
Answer. We apologize for this editing error in the article writing. A new sentences was incorporated as: “Then, it was added a solution (2% BSA) containing hERG1 (Alomone, Cat. APC-109), Hsp47 (MyBioSource, Cat. MBS9208399) as the primary antibody or the rabbit IgG (Alomone, RIC-001) for 30 min at RT. In addition, the anti-CD9 (BD Pharmingen, Cat. 555370), anti-CD63 (BD Pharmingen, Cat. 556019), anti-CD81 (BD Pharmingen, Cat. 555675), or the mouse IgG1 (BD Biosciences, Cat. 349040) for 30 min at RT also were tested. The immunolabeled particle-coupled beads were centrifugated at 8,000xg por 2 mins at 4°C. The pellets were washed, incubated with 10% BSA solution for 30 min at RT, centrifugated, and washed again with 1x PBS. The pellets were resuspended in a solution containing 2% BSA and secondary antibody a-mouse IgG1 Alexa Fluor 488 (BioLegend, Cat. 406626) or secondary antibody a-rabbit IgG Alexa Fluor 647 (BioLegend, Cat. 406414) for 30 min at RT. Finally, the sample was washed several times, and the pellet was resuspended in 1x PBS for acquisition in the cytometer CantoTM II cytometer (BD Biosciences, San Jose, CA, United States). The data acquired were analyzed using FlowJo software V10 (Tree Star, Ashland, OR, United States)[38].”.
- Line 274. “or the isotype control mouse IgG1 (BD Biosciences, Cat. 349040)”. Why mouse IgG? Since all the primary antibodies are rabbit antibodies, this control is meaningless for non-specific binding of rabbit Ig. Redo the experiments.
Answer. We apologize for this editing error in the article writing. A new sentences was incorporated as: “Then, it was added a solution (2% BSA) containing hERG1 (Alomone, Cat. APC-109), Hsp47 (MyBioSource, Cat. MBS9208399) as the primary antibody or the rabbit IgG (Alomone, RIC-001) for 30 min at RT. In addition, the anti-CD9 (BD Pharmingen, Cat. 555370), anti-CD63 (BD Pharmingen, Cat. 556019), anti-CD81 (BD Pharmingen, Cat. 555675), or the mouse IgG1 (BD Biosciences, Cat. 349040) for 30 min at RT also were tested. The immunolabeled particle-coupled beads were centrifugated at 8,000xg por 2 mins at 4°C. The pellets were washed, incubated with 10% BSA solution for 30 min at RT, centrifugated, and washed again with 1x PBS. The pellets were resuspended in a solution containing 2% BSA and secondary antibody a-mouse IgG1 Alexa Fluor 488 (BioLegend, Cat. 406626) or secondary antibody a-rabbit IgG Alexa Fluor 647 (BioLegend, Cat. 406414) for 30 min at RT. Finally, the sample was washed several times, and the pellet was resuspended in 1x PBS for acquisition in the cytometer CantoTM II cytometer (BD Biosciences, San Jose, CA, United States). The data acquired were analyzed using FlowJo software V10 (Tree Star, Ashland, OR, United States)[38]”.
- Lines 277-278. “secondary antibody a-mouse IgG1 Alexa Fluor 488 (BioLegend, Cat. 406626) for 30 min at RT.” The primary anti-hERG1 and anti-Hsp47 are rabbit antibodies. These experiments should not have worked. Using the proper animal combination (i.e. rabbit primary and anti-rabbit secondary), please redo the experiments.
Answer. We apologize for this editing error in the article writing. A new sentences was incorporated as “The pellets were resuspended in a solution containing 2% BSA and secondary antibody a-mouse IgG1 Alexa Fluor 488 (BioLegend, Cat. 406626) or secondary antibody a-rabbit IgG Alexa Fluor 647 (BioLegend, Cat. 406414) for 30 min at RT.”.
- Line 295. “1ng/µ”???
Answer. We apologize for this editing error in the article writing. A new sentences was incorporated as “1ng/µL”.
- Line 303. Change “spectrophotometer with an absorbance of 450 nm” to “spectrophotometer for an absorbance at 450 nm.”
Answer. According with reviewer´s suggestion the changes are included in the new version of the manuscript.
- Line 305. “The data are expressed as the mean ± SEM.” This may not be true as some data appear to be expressed as mean ± SD. Please check, and correct as necessary.
Answer. According with reviewer´s suggestion the changes are included in the new version of the manuscript.
- Line 309. Tittle. Change “hERG1 Is Expressed…” to “. hERG1 and Hsp47 are Expressed…” This change is necessary because Hsp47 expression is also described in this section. This change requires rewriting parts of this section. Please go over carefully so as to discuss both proteins.
Answer. According with reviewer´s suggestion the changes are included in the new version of the manuscript.
- Lines 312-313. “(ii) protein is expressed in the membrane of the extracellular vesicles, avoiding steps related to the lysis of sEVs.” In this sentence, the meaning of “avoiding steps related to the lysis of sEVs” is unclear. Delete this phrase.
Answer. According with reviewer´s suggestion the changes are included in the new version of the manuscript.
- Line 319. “hERG1 was present in the sEVs and associated with CD9 (a marker of sEVs).” This is a misleading statement. The data do not show that hERG1 is associated with CD9. In general, association of two molecules means binding of the two, hence this statement is an overinterpretation of the data. Change this sentence to “hERG1 was present in the sEVs as identified by CD9.”
Answer. According with reviewer´s suggestion the changes are included in the new version of the manuscript
- Line 324. Figure 1B. Here sEV membrane association of hERG1 (and later also Hsp47) is described. As I commented in my first review, this figure represents n=1 data. Such an illustration, no matter how nice looking it is, has little scientific value as such data could be selected or the labeling (or non-labeling) is accidental. This is a direct visualization that shows hERG1 and Hsp47 on sEV, and as such, it is a very important direct observation. However, showing only one example is not convincing. Please show several such cases. Also, it is good to show some statistical analyses (such as % of labeled sEVs out of >100 vesicles counted). The authors stated in their response that such demands cannot be readily met. If the presentation of these EM data cannot be scientifically improved, they should be deleted (as n=1 data has little scientific value), or they may be used as supplemental data. Please also see my comment #13.
Answer. We do not have data to run statistical analyses, such as % of labeled sEVs out of >100 vesicles counted. We decide to use this information only as a supplemental data as suggested by the reviewer.
- Line 335. Please describe how 13 positive and 13 negative cases were achieved? Please see my comments #7.
Answer. According with reviewer´s suggestion the changes are included in the new version of the manuscript, adding the next paragraph in the method section: “We enrolled 120 patients in the study, and 13 received a positive diagnosis of cardiac ischemia in the Stress Test by the cardiologists from Clínica Dávila. To have the same participants in both groups, 13 patients were randomly included with a negative diagnosis of cardiac ischemia in the Stress Test. The inclusion criterion for Stress Test included: age ≥ 18 years at the time of signing the informed consent.”
Regarding to requesting to include all the data we did not analyze the EV-hERG1 and EV-Hsp47 levels in the entire group. As a mentioned in the discussion section we are preparing an ELISA kit to run a study with a big number of participants to verify this preliminary data.
- Line 354. “Hsp47 was associated with CD9”. This is an overinterpretation of the data. The gel data does not show their binding.
Answer. According with reviewer´s suggestion the changes are included in the new version of the manuscript. A new sentence was wrote as: “Hsp47 was present in the sEVs as identified by CD9.”
- Line 368. “cardiomyocytes’”. Remove the apostrophe.
Answer. According with reviewer´s suggestion the changes are included in the new version of the manuscript
- Lines 372 and 374. Change “consists in” to “consists of”.
Answer. According with reviewer´s suggestion the changes are included in the new version of the manuscript
- Lines 388, 393 and 397. Please confirm if SD is correct as in the methods section, you state that (all) data are presented with SEM.
Answer. According with reviewer´s suggestion the changes are included in the new version of the manuscript. The SD is the correct state.
- Lines 389, 390, 394, and 398. “Significance is denoted as * p < 0.05 and ** p < 0.005 for normoxia.” Change “for” to either “compared with” or “vs”
Answer. According with reviewer´s suggestion the changes are included in the new version of the manuscript. The “vs” is the correct state.
- Lines 393 and 397. “n=3”. There are more than 3 data points. Please correct.
Answer. According with reviewer´s suggestion the changes are included in the new version of the manuscript
- Line 399. “Effect of Ischemia in Stress Test Participants on Secretion of sEVs…” This is a misleading title as the human study does not analyze secretion of sEVs. It looks at sEVs in blood, which is not secretion. Again, I see this as a subliminal suggestion that sEVs come from cardiomyocytes which appear to secrete sEVs under the ischemic condition in vitro. Some investigators believe and also have reported that AC16 cells are quite different from in site cardiomyocytes. In vitro data may not reflect what happens in vivo.
Answer. According with reviewer´s suggestion the changes are included in the new version of the manuscript.
The new title of the section 3.3 is “Blood levels of sEVs containing the hERG1 and Hsp47 Proteins in the Stress Test”.
Regardless to the “subliminal suggestion that sEVs come from cardiomyocytes” we have included in the discussion section of the new version the next discussion (line 734): “The in vitro data from cardiomyocyte cell culture suggests that the cell line releases EV-hERG1 and EV-Hsp47 to the extracellular media, and hypoxia increases these levels. However, considering our data, we cannot say that the EV-hERG1 and EV-Hsp47 found in blood samples are coming only from heart cardiomyocytes exposed to cardiac ischemia”.
- Line 439. Change “a secondary antibody conjugated with fluorescence” to “a fluorescent secondary antibody”.
Answer. According with reviewer´s suggestion the changes are included in the new version of the manuscript
- Lines 442-444. “In this manner, a small volume of PFP (5-10 mL) was used to obtain the total amount of 1x109 sEVs. The PFP samples were diluted to 0.5 mL with DPBS…” Delete “In this manner”. It is not clear how PFP in 5-10 mL can be diluted to make 0.5 mL of sample. Maybe something is not described. Please rewrite. Is 5-10 mL of PFP a small volume?
Answer. According with reviewer´s suggestion the changes are included in the new version of the manuscript: The “5-10 mL of PFP” was changed by “A small volume of PFP (5-100 mL) was used to obtain the total amount of 1x109 sEVs”.
- Lines 459-461. “Thus, we speculate that the EV-hERG1 and EV-Hsp47 concentrations in the blood samples increased, at least in the heart, in response to the development of ischemia.” What does “at least in the heart” mean? Was the blood sample collected from the heart? Please explain logically here what this phrase means. Unless there is a god explanation, please delete this phrase.
Answer. According with reviewer´s suggestion the changes are included in the new version of the manuscript. The phrase was delete.
- Lines 473-474. Are these SBP values shown with SD or SEM? How about the values in Table 2?
Answer. According with reviewer´s suggestion the changes are included in the new version of the manuscript. The SBP values shown with SD. Table 2.
- Figures 5, 6 and 7. In these studies when CHF and DHF are compared, it is not enough to compare just these two. The data from patients must be contrasted also to the data from healthy individuals. This is critical because, for example, it could be possible that the data from DHF patients are closer to those of the healthy individual! So, if diagnosis is one of the aims of this study, normal control must be included in the analysis.
Answer. According with reviewer´s suggestion the changes are included in the new version of the manuscript. We have included in the analysis a healthy group in the new figure 6. Please, see the description in the result section at: “3.5. Levels of sEVs Containing hERG-1 and Hsp47 Decreased in the Blood of Participants with Decompensated Heart Failure”.
The experiments in the figures 5 were done for characterize the presence of hERG1 and Hsp47 in the heart failure participants, so we do not believe necessary include the healthy group. In the figure 7, because the aim was to identify an algorithms to evaluate the transition for CHF to DHF. Again we believe is not necessary to include the healthy group.
- Lines 529-531. “we analyzed the correlations between sEVs/EV-hERG1 and sEVs/EV-Hsp47 and the total concentrations of EV and EV-Hsp47 for each group of patients.” It is not clear what this sentence means. The figure legend seems to describe the analyses better.
Answer. According with reviewer´s suggestion the changes are included in the new version of the manuscript. The new sentence is: “Given these results, we analyzed the concentrations of sEVs, EV-hERG1, and EV-Hsp47 as an algorithm to discriminate between compensated and decompensated heart failure. We study the correlation between (i) sEVs and EV-hERG1 and (ii) sEVs and EV-Hsp47 (EV-Hsp47/mL) correlation curves for compensated heart failure (Figure 7A). In addition, the correlation between (i) sEVs and EV-hERG1 and (ii) sEVs and EV-Hsp47 correlation curves were done for decompensated heart failure (Figure 7B).”.
- Figure 7. The appearance of the two lines in A and B are the same. Please make the two lines distinct (by color, one solid and one dotted, etc.) and indicate what is what.
Answer. According with reviewer´s suggestion the changes are included in the new version of the manuscript.
- Lines 627-628. “the secretion of hERG1 by extracellular vesicles could be a response mechanism against cardiac damage.” The meaning of this sentence is not clear. Is hERG1 secreted by EVs?
Answer. According with reviewer´s suggestion the changes are included in the new version of the manuscript. The new sentence is: “Here, we report for the first time the presence of hERG1 in the membrane of extracellular vesicles in peripheral blood samples, using an anti-hERG1 antibody that recognizes the region between the S1 and S2 domains of hERG1, suggesting that the presence of hERG1 in the surface of the extracellular vesicles could be associated with cardiac damage.”.
- Line 636. “Hsp47 myofibroblasts are the primary mediators of tissue fibrosis…” Do you mean “Hsp47 in myofibroblasts is the primary mediator of tissue fibrosis…”?
Answer. According with reviewer´s suggestion the changes are included in the new version of the manuscript. The sentences was changed to “Hsp47 in myofibroblasts is the primary mediator of tissue fibrosis”
- Lines 659-660. “signaling to the heart the cell type involved in the secretion of sEVs containing hERG1 and Hsp47.” The meaning of this phrase is not clear. Please rewrite.
Answer. According with reviewer´s suggestion the changes are included in the new version of the manuscript. The phrase was deleted in the new version of the manuscript.
- Figure 8. The cartoon of the EV shows Hsp47 is embedded in the membrane of the vesicle. As far as I remember, Hsp47 does not have a transmembrane domain. How does it span the membrane as depicted in the figure? Please discuss how this protein can be expressed on the outer surface of sEV. It may be necessary to modify this cartoon.
Answer. We considered the reviewer´s comment and we included a new sentence in the discussion section of the manuscript (Line 1073): “Hsp47 is an intracellular chaperone protein that is vital for collagen biosynthesis. In addition, Hsp47 is present on the surface of platelets, having an extracellular role, where it interacts with collagen, stabilizing platelet adhesion and thrombus formation. The ability of chaperone proteins to function in the extracellular environment may be important in a number of pathophysiological circumstances. It is unclear how Hsp47 exerts this effect, whether through modulation of ligand structure or it functions as an adhesion protein to enhance platelet–collagen interactions[55]. Using a commercial antibody against “TKDVERTDGAL” sequence of human Hsp47, we found a positive signal for Hsp47 by Flow Cytometer (Figures 1 and 5) and NTA (Figures 3 and 5) in sEVs from human platelet-free plasma. In addition, we observed Hsp47 in the sEVs surface from the cell line of human cardiomyocyte (Figure 2). This evidence suggests that Hsp47 are binding in unknow way to the surface of sEVs (Figure 8).".
The cartoon was modified, according with the reviewer´s suggestion. The figure legend is: “Figure 8. Potential biomarkers of cardiovascular diseases (CVDs). We found small extracellular vesicles (i.e., EVs sized from 50 to 200nm), containing hERG1 and Hsp47 proteins in the surface of sEVs from blood samples. All this evidence suggest that hERG1 and Hsp47 are binding in unknow way to the surface of sEVs. Our finding suggests that vesicular hERG1 and Hsp47 might be explored as possible biomarkers of CVDs”.
Reviewer 3 Report
Comments and Suggestions for Authors
I'm really sorry, but all my requests on EV characterization must be addressed.
This is mandatory working on EVs
Author Response
I'm really sorry, but all my requests on EV characterization must be addressed. This is mandatory working on EVs
Answer. We understand that the suggested changes are necessary to produce a better manuscript. In the new document, all the requests on EV characterization are incorporated with the reliable data. Please see the updated version of the manuscript for your revision and consideration to publish.
Round 3
Reviewer 2 Report
Comments and Suggestions for Authors
The authors have made an effort to revise the manuscript. Although one can ask for more work, I feel that the paper clears the scientific standard of the journal. However, writing is still muddy, and I have made suggestions for making the text a bit more readable. There is a huge room to improve writing, but I think that readers can get the correct message from the current text once my suggested modifications are accepted. If my modification changed the meaning of a sentence, please rewrite it. Since Abstract is the most read part of the paper, I have made a substantial number of changes.
1. Abstract. I believe that the following changes are necessary to make a readable abstract. I ask the authors to examine and consider these edits.
Line 21. Change “characterized the presence of hERG1 and Hsp47 in the surface of small extracellular vesicles (sEVs) from blood samples and their association with CVD” to “demonstrates the presence of human ether-à-go-go-related gene 1 (hERG1) and heat shock protein 47 (Hsp47) on the surface of small extracellular vesicles (sEVs) in human peripheral blood and their association with CVD.”
Line 23. Change “26 participating in a stress test were enrolled” to “26 participants in a cardiac stress test were enrolled.”
Line 26. Change “exposing to the extravesicular domain the sequences 430AFLLKETEEGPPATE445 for hERG1 and 169ALQSINEWAAQTT- DGKLPEVTKDVERTD196 for Hsp47” to “exposing extravesicularly the sequences 430AFLLKETEEGPPATE445 for hERG1 and 169ALQSINEWAAQTT- DGKLPEVTKDVERTD196 for Hsp47.”
Line 29. Change “into the extracellular media, and human cardiomyocytes released sEVs containing hERG1 (EV-hERG1) and/or Hsp47 (EV-Hsp47)” to “into the media, and human cardiomyocytes in culture also released sEVs containing hERG1 (EV-hERG1) and/or Hsp47 (EV-Hsp47).“
Line 32. Change “the levels of EV-hERG1 and EV-Hsp47 in the blood decreased with decompensated heart failure (DHF)” to “the plasma levels of EV-hERG1 and EV-Hsp47 decreased in patients with decompensated heart failure (DHF).”
Line 34. Change “and they were also found in peripheral blood” to “and also in those isolated from human peripheral blood.”
Line 35. Change “EV-Hsp47 can be…” to “EV-Hsp47 may be…”
2. Line 98. Change back to “Thus”. “In consequence” does not make sense here.
3. Line 129. Change “To have the same participants…” to “To have the same number of participants…”
4. Line 131. Change “Stress Test included…” to “Stress Test was…”
5. Line 224: Change “Then, a PFP aliquot was used...” to “An aliquot of FPF was used…”
6. Line 225. Change “The diluted samples were incubated for 1 hour at 37ºC with primary antibodies. The anti-rabbit IgGantibodies used were either anti-hERG1 (APC-109, Alomone labs) or anti-HSP47 (MBS9208399, MyBioSource), which recognize the 430AFLLKETEEGPPATE445 and 169ALQSINEWAAQTTDGKLPEVTKDVERTD196 sequences, respectively” to “The diluted samples were incubated for 1 hour at 37ºC with either rabbit anti-hERG1 (APC-109, Alomone labs) or rabbit anti-HSP47 (MBS9208399, MyBioSource), which recognizes the 430AFLLKETEEGPPATE445 or 169 ALQSINEWAAQTTDGKLPEVTKDVERTD196 sequences, respectively.”
7. Line 265. “Briefly, the pool of sEVs purified by ultracentrigugation were fixed (4% formaldehyde).” Important information is missing. Please describe how long samples were fixed at what temperature. Also provide information on the solution in which 4% formaldehyde was made.
8. Line 270. Change “Uranil” to “uranyl acetate”.
9. Line 294. Change “Then, it was added a solution (2% BSA) containing hERG1 (Alomone, Cat. APC-109), Hsp47 (MyBioSource, Cat. MBS9208399) as the primary antibody or the rabbit IgG (Alomone, RIC-001) for 30 min at RT” to “Then, the samples were mixed with a 2% BSA solution containing anti-hERG1 (Alomone, Cat. APC-109), anti-Hsp47 (MyBioSource, Cat. MBS9208399) as the primary antibodies or rabbit IgG (Alomone, RIC-001).”
10. Line 339. Title. Change “in the surface” to “on the surface”. It is OK to say in the membrane, but it has to be on the surface (not in the surface).
11. Line 342. Change “from human samples,” to “from human peripheral blood.” End the sentence here.
12. Line 342. Change “considering they are (i) highly expressed in the heart and (ii) protein is associated with the membrane of the extracellular vesicles” to “We found that some of them were highly expressed in the heart and that (ii) they were associated with the membrane of the small extracellular vesicles.”
13. Line 344. Delete “Thus”.
14. Line 357. Change “in the pellet from 12,000xg (free-sEVs fraction) when the anti-hERG1 antibody was included in the reaction” to “in the 12,000xg centrifugation pellet (EV-free fraction) even when the anti-hERG1 antibody was included in the reaction”.
15. Line 359. Change “positive signal in the pellet from 110,000xg (sEVs enriched fraction) were” to “a positive signal in the 110,000xg centrifugation pellet (sEVs enriched fraction) was”.
16. Line 367. Change “electron microscopy (TEM) was also used to characterize hERG1 signal associated to sEVs” to “electron microscopy (TEM) was also used to show hERG1 association with sEVs”.
17. Line 368. Change “that hERG1 is present in the membrane surface of sEVs obtained from peripheral blood samples, and the sequence is apparently exposed to the extravesicular region” to “that the 430AFLLKETEEGPAPATE445 epitope of hERG1 is present on the surface of sEVs obtained from human peripheral blood”.
18. Line 389. Change “Hsp47 is expressed in the surface of extracellular vesicles” to “Hsp47 is also expressed on the surface of extracellular vesicles.” Do not italicize this sentence.
19. Line 390. Change “…protein 70 (Hsp70) is expressed in the…” to “…protein 70 (Hsp70) was expressed on the…”.
20. Line 400. Change “a negative signal for free-sEVs pellet (12,000xg; Figure 1B) and a positive signal for the whole sEVs (110,000xg; Figure 1D)” to “a negative signal for the sEV-free 12,000xg centrifugation pellet (Figure 1B) but a positive signal for the 110,000xg centrifugation sEV pellet (Figure 1D)”.
21. Line 402. Change “is associated with the extravesicular domain of sEVs” to “is exposed to the extravesicular domain of sEVs”.
22. Line 403. Change “was used to characterize the Hsp47 signal from” to “was used to indicate EV surface localization of the Hsp47 epitope (ALQSINEWAAQTTDGKLPEVTKDVERTD).
23. Line 485. Change “that that uses an anti-hERG1” to ”with either anti-hERG1”.
24. Line 574. Change “CHF compared to healthy but” to “CHF compared with healthy individuals but”.
25. Line 581. Authors state, “Given these results, we analyzed the concentrations of sEVs, EV-hERG1, and EV-Hsp47 as an algorithm to discriminate between compensated and decompensated heart failure.” This statement carries little power of persuasion because the two types of heart failure can be cleanly discriminated (as the authors did) using the criteria outlined in the Methods. So, instead of rationalizing the study in this unconvincing manner, authors should consider saying that sEVs can be used to further characterize pathophysiology of CHF and DHF.
26. Line 583. “We study the correlation between (i) sEVs and EV-hERG1 and (ii) sEVs and EV-Hsp47 (EV-Hsp47/mL) correlation curves for compensated heart failure (Figure 7A). In addition, the correlation between (i) sEVs and EV-hERG1 and (ii) sEVs and EV-Hsp47 correlation curves were done for decompensated heart failure”. This is very confusing. Consider the following: “To this end, we studied the correlation between (i) total sEVs and EV-hERG1 and (ii) total sEVs and EV-Hsp47 for both compensated as well as decompensated heart failures (Figure 7).“
27. Line 677. “In another study in patients with heart failure (HF), with dilated cardiomyopathy (DCM) and ischemic cardiomyopathy (ICM) as an end-stage of HF in humans, were not found changes in the hERG1 mRNA of cardiac tissue.” The meaning of this sentence is unclear. Try, “In another study on heart failure (HF) patients with dilated cardiomyopathy (DCM) vs ischemic cardiomyopathy (ICM) as end-stages, there was no difference in hERG1 mRNA levels within the cardiac tissue.”
28. Linr 687. Change “suggesting that the presence of hERG1 in the surface of the extracellular vesicles could be associated with cardiac damage” to “suggesting that the presence of hERG1 on the surface of the extracellular vesicles may reflect damaged conditions of the cardiac tissue.”
29. Lines 708, 709 and 732. Change “in the surface of sEVs” to “on the surface of sEVs”.
30. Line 726. Change “in the sEVs surface” to “on the sEVs surface”.
31. Line 748. Change “heart failure and healthy” to “heart failure and healthy individuals”.
32. Line 769. Change “that extracellular vesicles containing in their sur-face hERG1 (sVE-hERG1) and Hsp47 (sVE-Hsp47) proteins, increasing as a result of exposure to hypoxia in human cardiomyocytes cell line and in blood samples during cardiac ischemia induced by a stress test” to “that small extracellular vesicles secreted by cultured human cardiomyocytes as well as those present in circulating blood express on their surface hERG1 and Hsp47 proteins. When the cardiomyocytes were exposed to hypoxia or when cardiac ischemia was induced in humans, the secreted and circulating sEV levels increased.”
33. Figure legends.
Line 374. Change the title to “The hERG1 (EV-hERG1) and Hsp47 (EV-Hsp47) epitopes are expressed on the surface of sEVs from cardiac ischemia patients.”
Line 375. Change “A pool samples from positive cardiac ischemia participants…. (B)” to “(A) A pooled sEV samples from individuals with cardiac ischemia were purified from platelet-free plasma by several ultracentrifugation steps and tested by Western blotting using anti-hERG1, anti-Hsp47 and anti-CD9 antibodies. Both the sEV-free fraction (-, the 12,000 xg pellet) and the sEV enriched fraction (+, 110,000 xg pellet) were analyzed. (B)”
Line 598. Change “(DHF, n=10) was used…” to “(DHF, n=10) participants was used...”
Line 780. Change “in its surface” to “on its surface.”
Comments on the Quality of English LanguageCheck for punctuations and typos.
Author Response
Dear reviewer, we would like to thank you for the excellent and generous work you have done in reviewing our manuscript. Without a doubt, your contribution has allowed the manuscript to achieve an adequate level in aspects related to scientific information, as well as in the correct communication of the results.
In the new version of the manuscript all the suggestions that you have proposed in the third revision have been incorporated. Please review the changes made with tracked changes.
Reviewer 3 Report
Comments and Suggestions for Authors
The Ms still lacks the negative EV marker. This is mandatory
Comments on the Quality of English LanguageSome grammar errors must be corrected
Author Response
Dear reviewer, thank you very much for your new response.
In the second version of the manuscript we had included (Supplementary Figure 1) an experiment in which, using extracellular vesicles purified by ultracentrifugation from the peripheral blood of participants with CVDs, the absence of Calnexin expression by Western blot.
In this new version of the manuscript we have included a Western blot of Grp94, as a new negative marker of the sEVs. In order to have better evidence of negative markers of sEVs.
Calnexin and GRP94, are considered as an intracellular markers and negative for sEVs isolated from blood plasma (1).
In the publication "Minimal information for studies of extracellular vesicles (MISEV2023): From basic to advanced approaches", cited as reference 59 of the new version of the article, it is noted that: "Because of the heterogeneity of EVs, MISEV2023, like MISEV2018, cannot recommend molecular markers of specific EV subtypes. MISEV2023 recommends the five-component framework introduced in MISEV2018 for reporting claims about the protein content of EVs. Categories 1 and 2 assess the presence of EVs features. Category 3 assesses purity from common contaminants. Categories 4 and 5 provide additional information on possible intracellular origins of EVs (4) or co-isolates (5). Ideally, enrichment or depletion of markers in EV preparations versus unfractionated source material should be shown. To avoid perceived restrictions on which EV proteins should be analyzed, MISEV2023 gives only a few nominative examples".
In the above publication, CD9, CD63, and CD81 are considered as EVs features. On the other hand, calnexin and Grp94 are mentioned in category 4 as proteins of intracellular origin and not as proteins that characterize sEVs as categories 1 and 2.
Finally, as the MISEV2023 publication points out, the positive and negative markers of sEVs are under review and the MISEV2023, "cannot recommend molecular markers of specific EV subtypes.".
In this way, our data suggest that the hERG1 and Hsp47 proteins are associated with CD9, CD63, and CD81 and not to Calnexin or Grp94.
Reference:
1. Hong, C.S., Diergaarde, B. & Whiteside, T.L. Small extracellular vesicles in plasma carry luminal cytokines that remain undetectable by antibody-based assays in cancer patients and healthy donors. BJC Rep 2, 16 (2024). https://doi.org/10.1038/s44276-024-00037-x
Round 4
Reviewer 3 Report
Comments and Suggestions for Authors
The authors addressed my requests